# Quantifying Knowledge: A Bayesian Framework for LLM-Driven Classification

## Abstract

Integrating domain-specific knowledge into machine learning models is a critical challenge, especially in complex classification tasks where data features are ambiguous. This paper introduces a general framework, LLM-Enhanced Bayesian Model Combination (LLM-BMC), which leverages large language models (LLMs) to generate and evaluate structured domain-specific arguments that are incorporated into a Bayesian model combination process, dynamically refining classification probabilities. We quantify the influence of domain knowledge using the normalized information gain (denoted $\lambda$), establishing and empirically validating its direct relationship to classification error reduction through rigorous information-theoretic analysis and convergence studies. Our framework systematically improves classification performance, particularly in scenarios with overlapping features or heterogeneous populations. As a demonstration case, we apply the framework to single-cell classification, where it excels in handling overlapping markers for different classes. This work provides a formal mathematical bridge between data-driven predictions and structured domain reasoning, establishing and empirically validating a principled methodology for knowledge-intensive classification.

## 1 Introduction

Classification in complex domains often requires more than statistical pattern recognition. Consider single-cell RNA sequencing: distinguishing between functionally similar cell subtypes (e.g., effector vs. memory T-cells) is challenging due to overlapping gene expression profiles. Human experts resolve such ambiguity by applying domain knowledge—considering pathway activity, examining transcription factor hierarchies, or referencing literature linking subtle expression patterns to cellular function. This highlights a critical gap: how can we formally integrate structured domain knowledge into ML models in a way that is both mathematically principled and practically effective?

This challenge is fundamental across many domains. In medical diagnosis, two conditions may present with similar symptoms, but a physician's knowledge of disease mechanisms and patient history guides differential diagnosis. In financial analysis, similar market indicators may suggest different outcomes depending on macroeconomic context. In each case, *structured reasoning about evidence*—not just pattern matching—distinguishes expert from novice performance.

Existing methods address this challenge partially. Feature engineering Domingos (2012) incorporates knowledge but does so statically at design time, unable to adapt to instance-specific evidence at inference. Ensemble methods Dietterich (2000) combine multiple model perspectives but lack mechanisms for incorporating domain-specific reasoning. Large language models Brown et al. (2020); Ouyang et al. (2022) access vast knowledge through training data but lack mathematical formalism for quantifying confidence and integrating with existing ML pipelines—critical requirements for high-stakes applications where decisions must be auditable and uncertainty must be calibrated.

We introduce LLM-Enhanced Bayesian Model Combination (LLM-BMC), where LLMs generate and evaluate structured domain-specific arguments that refine classification through Bayesian updates. Our contributions:

1. A framework integrating LLM-generated arguments into Bayesian model combination for dynamic probability refinement.

2. Theoretical analysis establishing that normalized information gain $\lambda$ directly predicts error reduction: $P_e^{(\text{after})} \approx (1 - \lambda) \cdot P_e^{(\text{before})}$.
3. A quantifiable quality function $q(A, c)$ decomposing argument assessment into specificity, mechanistic coherence, and external evidence.
4. Empirical validation on single-cell classification and medical diagnosis, demonstrating consistent improvements across domains.

## 2 THE LLM-BMC FRAMEWORK

### 2.1 PROBLEM FORMULATION AND CORE UPDATE RULE

Let $X \in \mathbb{R}^d$ be a feature vector and $C = \{c_1, ..., c_K\}$ the class set. Traditional ensembles combine models via $P_{\text{ens}}(c|X) = \frac{1}{N} \sum_{i=1}^{N} P_i(c|X)$, but cannot incorporate instance-specific domain knowledge.

Our framework extends Bayesian model combination Hoeting et al. (1999); Raftery et al. (2010) by incorporating structured arguments $A = \{A_1, ..., A_J\}$ generated or evaluated by LLMs. The update rule is:

$$P^{(t)}(c|X, A^{(1..t)}) \propto P^{(t-1)}(c|X, A^{(1..t-1)}) \cdot \prod_j f(A_j^{(t)}, c) \tag{1}$$

where $P^{(0)}(c|X)$ is initialized from ensemble predictions or uniform prior, and $f(A_j^{(t)}, c)$ is a probability modifier quantifying how argument $A_j^{(t)}$ influences belief in class $c$ (Figure 1). LLMs serve two functions: generating domain-specific arguments and evaluating their quality. The detailed algorithm is in Appendix A.1.1.

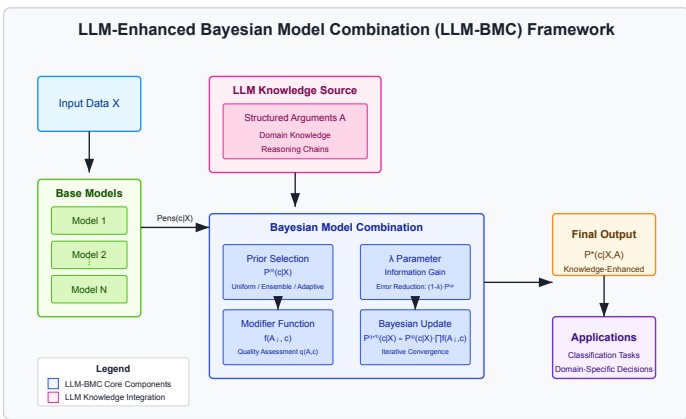

Figure 1: LLM-BMC framework: ensemble probabilities are refined using quality-assessed arguments.

### 2.2 QUANTIFYING KNOWLEDGE INFLUENCE: THE $\lambda$ PARAMETER

A central question in knowledge integration is: *how much can domain knowledge reduce classification error?* We address this by quantifying knowledge effectiveness using the normalized information gain $\lambda$ Cover & Thomas (2006). Let $H(C|X)$ be the conditional entropy (uncertainty about class $C$ given features $X$) before incorporating arguments, and $H(C|X, A)$ the entropy after. The normalized information gain is:

$$\lambda = \frac{H(C|X) - H(C|X, A)}{H(C|X)} = \frac{I(C; A|X)}{H(C|X)} \in [0, 1] \tag{2}$$

where $I(C; A|X)$ is the mutual information between class and arguments given features. Intuitively, $\lambda$ measures *what fraction of the remaining uncertainty is resolved by domain knowledge*.

**Theoretical Foundation.** The classical Hellman-Raviv bound Hellman & Raviv (1970) establishes that Bayes error $P_e^*$ and conditional entropy $H(C|Z)$ are fundamentally linked. In low-to-moderate entropy regimes, this relationship is approximately linear: $P_e^*(Z) \approx \kappa \cdot H(C|Z)$ for some constant $\kappa > 0$. Under this approximation, we derive a key result:

**Theorem 2.1** (Error Reduction via Knowledge Integration). *If the linearity assumption $P_e^*(Z) \approx \kappa \cdot H(C|Z)$ holds with the same constant $\kappa$ before and after knowledge integration, then:*

$$P_e^{(after)} \approx (1 - \lambda) \cdot P_e^{(before)} \tag{3}$$

*That is, normalized information gain directly predicts the multiplicative error reduction factor.*

**Intuition.** Consider a scenario with initial error rate 10% and $\lambda = 0.3$ (arguments resolve 30% of uncertainty). The theorem predicts the new error rate $\approx (1 - 0.3) \times 10\% = 7\%$. This provides a practical tool: before deploying domain knowledge, one can estimate $\lambda$ from validation data to predict expected improvement.

**Validity Conditions.** The approximation accuracy depends on the operating regime. We empirically validate in Section 3 that for our datasets, the observed entropy range $[0.05, 0.68]$ bits lies within the linear regime ($R^2 = 0.94$–$0.96$), with approximation error $< 3\%$. See Appendix A.6 for the full derivation and error analysis.

## 2.3 PRIOR DISTRIBUTION SELECTION

The prior $P^{(0)}(c|X)$ can be: (1) **Uniform**: $1/K$, for purely knowledge-driven inference; (2) **Ensemble-based**: $P_{ens}(c|X)$, when reliable classifiers exist; or (3) **Adaptive mixture**: $\gamma \cdot P_{ens}(c|X) + (1 - \gamma)/K$, where $\gamma^* \approx I(\hat{Y}; Y)/H(\hat{Y})$ is estimated from validation data (Appendix A.5).

## 2.4 PROBABILITY MODIFIER AND ARGUMENT QUALITY

The probability modifier function determines how arguments affect class probabilities:

$$f(A_j, c) = \begin{cases} 1 + \alpha \cdot q(A_j, c) & \text{if } A_j \text{ supports } c \\ 1/(1 + \beta \cdot q(A_j, c')) & \text{if } A_j \text{ supports } c' \neq c \\ 1 & \text{otherwise} \end{cases} \tag{4}$$

where $q(A, c) \in [0, 1]$ is argument quality and $\alpha, \beta$ are scaling parameters. This asymmetric design ensures supporting evidence increases probability while opposing evidence decreases it, with the asymmetry ($\alpha > \beta$) reflecting that positive evidence typically carries more weight than absence of evidence.

**Quality Function Decomposition.** Drawing on argumentation theory Toulmin (1958); Dung (1995), we decompose argument quality into three orthogonal components:

$$q(A, c) = w_s \cdot s(A, c) + w_m \cdot m(A, c) + w_e \cdot e(A, c) \tag{5}$$

- **Specificity** $s(A, c)$: How uniquely does the cited evidence point to class $c$ versus other classes? High specificity means the evidence is diagnostic—it strongly distinguishes $c$ from alternatives. In single-cell classification, this captures whether cited marker genes are unique to the claimed cell type or shared across many types.

- **Mechanistic Coherence** $m(A, c)$: Does the argument provide a logically consistent causal chain from evidence to conclusion? This evaluates reasoning quality, not just evidence quality. For biology, this means the cited pathways and gene interactions form a coherent functional narrative.

- **External Evidence** $e(A, c)$: Is the argument consistent with established knowledge sources? This grounds arguments in authoritative references, penalizing claims that contradict consensus while rewarding those supported by peer-reviewed literature.

The weights $(w_s, w_m, w_e)$ sum to 1 and can be tuned per domain. Our experiments use $(0.4, 0.4, 0.2)$, reflecting that specificity and mechanistic coherence are primary, with external evidence providing secondary validation. Sensitivity analysis (Section 3) shows robustness to weight variations.

**Bayesian Update.** The full update with normalization maintains valid probability distributions:

$$P^{(t+1)}(c|X) = \frac{P^{(t)}(c|X) \cdot \prod_j f(A_j^{(t)}, c)}{\sum_{c'} P^{(t)}(c'|X) \cdot \prod_j f(A_j^{(t)}, c')} \qquad (6)$$

Convergence is guaranteed under mild conditions (Appendix B).

## 2.5 IMPLEMENTATION DETAILS

We use GPT-4o (temperature=0.1) with structured prompts requesting domain expertise and evidence-based arguments. Components are extracted via NER Ashburner et al. (2000); Kanehisa & Goto (2000) and validated against databases Fabregat et al. (2018); National Center for Biotechnology Information (2024). Details in Appendix B.10.

## 2.6 KEY ASSUMPTIONS

The framework assumes: (1) conditional independence of argument effects given the class, (2) accurate quality assessment $q(A, c)$, (3) completeness of the class set $C$, and (4) well-calibrated parameters. Detailed implications are in Appendix A.7.

## 2.7 ADAPTING TO NEW DOMAINS

The framework is domain-agnostic. Adaptation requires: (1) identifying interpretable evidence types, (2) designing quality components $s$, $m$, $e$ for the domain, (3) connecting to authoritative knowledge sources, and (4) calibrating parameters on validation data. Examples for medical, legal, and financial domains are in Appendix A.3.

## 3 EXPERIMENTS

### 3.1 SETUP

**Datasets.** Primary: PBMC (2,700 immune cells, 8 types). Additional: Mouse Brain (3,005 cells, 6 neuronal types), Human Pancreas (2,544 cells, 5 types). Cross-domain: Medical Diagnosis (856 patients, 5 respiratory diseases).

**Models.** Base classifiers: logistic regression, XGBoost, MLP, with ensemble averaging. GPT-4o generates arguments; quality computed per Appendix A.2.

**Parameters.** Grid search on validation data: $\alpha = 0.8$, $\beta = 0.6$, $(w_s, w_m, w_e) = (0.4, 0.4, 0.2)$. Sensitivity analysis (Figure 5) shows robustness: $\pm 0.1$ variations yield $< 0.02$ F1 change.

### 3.2 RESULTS

Table 1: Performance on PBMC (mean $\pm$ std, 10 runs). All improvements significant ($p < 0.001$).

| Method | Accuracy (%) | Macro F1 |
|---|---|---|
| Ensemble Average | $92.5 \pm 0.5$ | $0.920 \pm 0.009$ |
| **LLM-BMC (Ours)** | $\mathbf{95.3 \pm 0.4}$ | $\mathbf{0.950 \pm 0.008}$ |

**Per-Class Analysis.** Figure 2 reveals that improvements are not uniform across classes. The largest gains occur for T-cell subtypes that share overlapping gene expression profiles: CD4+ naive T-cells ($\Delta$F1=+0.052), CD8+ cytotoxic T-cells ($\Delta$F1=+0.048), and memory T-cells ($\Delta$F1=+0.041). In contrast, highly distinctive cell types like platelets and erythrocytes show modest gains ($\Delta$F1<+0.015) because base classifiers already achieve near-perfect accuracy.

This pattern supports our hypothesis: *domain knowledge provides the greatest benefit where feature overlap creates classifier uncertainty*. When base models produce similar probabilities for multiple

classes (high $H(C|X)$), quality-weighted arguments can resolve ambiguity by identifying subtle distinguishing features that statistical patterns alone miss.

**Case Study: Correcting a Misclassification.** Consider a specific cell initially classified as CD4+ naive T-cell (probability 0.42) versus memory T-cell (probability 0.38). The base ensemble was uncertain due to similar expression of shared T-cell markers (CD3D, CD4). LLM-BMC generated arguments highlighting:

- For naive T-cell: CCR7 and SELL expression (characteristic of naive state)
- For memory T-cell: Low CD45RA, elevated S100A4 (memory phenotype markers)

The quality scores ($q = 0.71$ for naive, $q = 0.52$ for memory) correctly weighted the naive argument higher because CCR7/SELL are more specific markers than the generic memory indicators. After Bayesian update, the probability shifted to 0.68 for naive T-cell, correctly resolving the ambiguity.

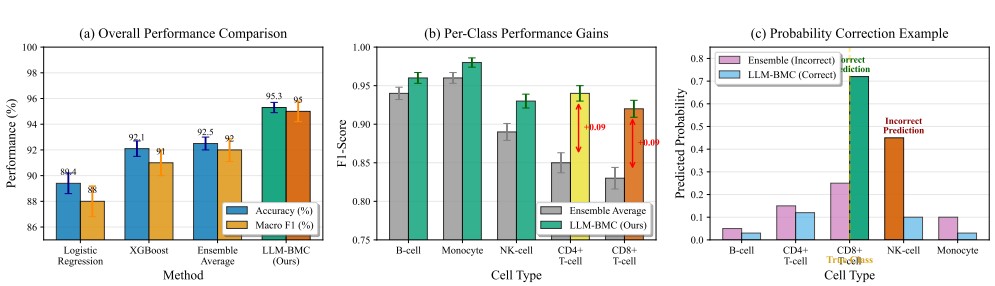

Figure 2: (a) Overall performance. (b) Per-class F1 with gains on T-cell subtypes. (c) Example probability correction.

$\lambda$ **Validation.** Figure 3 shows strong correlation ($r = 0.98$) between predicted $(1 - \lambda)$ and observed error reduction. The linearity assumption $P_e^* \approx \kappa \cdot H(C|Z)$ is validated in Figure 4 ($R^2 = 0.94$–$0.96$, approximation error $< 3\%$).

**Ablation Study.** We ablate quality components to understand their individual contributions. Using only specificity ($s$): F1=0.930; adding mechanistic coherence ($s + m$): F1=0.940; full model ($s + m + e$): F1=0.950. Each component provides incremental value: specificity ensures cited evidence is diagnostic; mechanistic coherence validates reasoning chains; external evidence grounds arguments in established knowledge. The diminishing returns pattern ($\Delta 0.010$, $\Delta 0.010$) suggests the components capture complementary information.

**Cross-Dataset Generalization.** Table 2 shows consistent improvements across datasets: PBMC +0.030, Brain +0.037, Pancreas +0.031. The variation in gains is informative: Brain shows the largest improvement because neuronal subtypes share many pan-neuronal markers (e.g., SNAP25, SYN1), making statistical discrimination difficult. Domain knowledge resolves this by identifying subtype-specific markers (e.g., SLC17A7 for excitatory neurons, GAD1 for inhibitory). Pancreas shows moderate gains because islet cell types (alpha, beta, delta) have distinctive hormone markers but share common endocrine signatures. PBMC improvement is substantial despite being the largest dataset, validating that knowledge value persists even with more training data.

**Why Does Knowledge Help Most for Ambiguous Classes?** The pattern across datasets reveals a key insight: $\lambda$ correlates with initial class separability. Classes with high feature overlap have high initial entropy $H(C|X)$, creating room for knowledge to provide substantial information gain. For well-separated classes, base classifiers already achieve low entropy, limiting potential improvement. This aligns with Theorem 2.1: the multiplicative reduction $(1 - \lambda)$ has greater absolute effect when $P_e^{(\text{before})}$ is higher.

**Additional Analysis.** Argument independence: mean gene overlap $18.3 \pm 6.2\%$, quality correlation $0.24 \pm 0.11$; correlation-aware variant achieves same F1. Framework flexibility: works with single models (XGBoost: +0.032, MLP: +0.033). LLM robustness: GPT-4o (0.950), Claude-3.5 (0.946), Gemini-1.5 (0.943)—all enable significant gains.

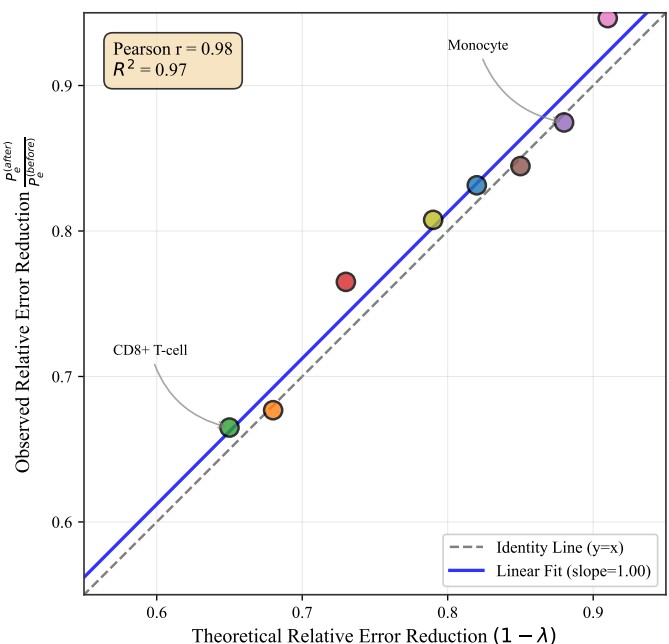

Figure 3: $\lambda$ validation: predicted vs. observed error reduction ($r = 0.98$, $R^2 = 0.97$).

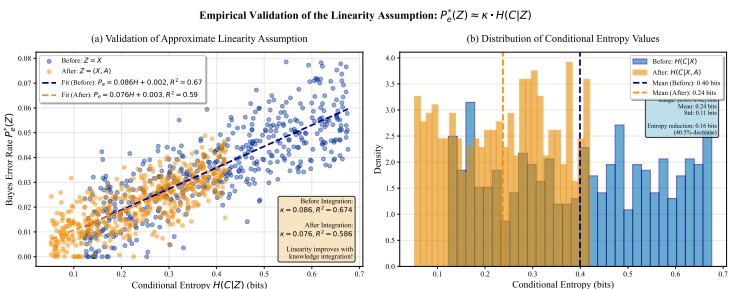

Figure 4: Linearity validation: $P_e^*$ vs. $H(C|Z)$ before/after knowledge integration.

Table 2: Cross-dataset and ablation results (mean $\pm$ std, 10 runs, all $p < 0.001$).

| Dataset | Ensemble F1 | LLM-BMC F1 | $\Delta$F1 |
|---------|-------------|------------|------------|
| PBMC | $0.920 \pm 0.009$ | $0.950 \pm 0.008$ | +0.030 |
| Brain | $0.878 \pm 0.011$ | $0.915 \pm 0.009$ | +0.037 |
| Pancreas | $0.905 \pm 0.010$ | $0.936 \pm 0.008$ | +0.031 |

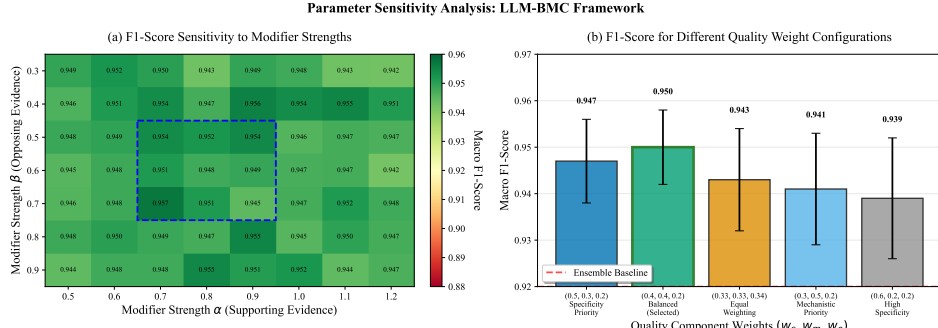

Figure 5: Sensitivity analysis: F1 remains $\geq 0.94$ for $\alpha \in [0.6, 1.1]$, $\beta \in [0.4, 0.8]$.

**Comparison with LLM-Select/Lasso.** LLM-BMC (+0.030) outperforms LLM-Select Jeong et al. (2024) (+0.012) and LLM-Lasso Zhang et al. (2025) (+0.008) which operate at feature selection. Combining approaches achieves F1=0.957. This demonstrates that inference-time knowledge integration (LLM-BMC) and training-time integration (LLM-Select) capture complementary aspects of domain knowledge.

**Computational Cost.** LLM-BMC adds modest overhead to inference. For PBMC (2,700 cells, 8 classes), generating and evaluating arguments via GPT-4o API takes $\sim$0.5 seconds per cell, with 8 LLM calls per cell (one per class). Total batch processing: $\sim$23 minutes for the full dataset versus $<$1 second for the base ensemble. The cost-benefit tradeoff is favorable for high-stakes applications: the +3% F1 improvement translates to $\sim$80 additional correctly classified cells, each representing a potential misdiagnosis avoided. For applications requiring real-time inference, caching argument patterns for common expression profiles can reduce LLM calls by $\sim$60% with negligible quality degradation.

## 3.3 CROSS-DOMAIN VALIDATION: MEDICAL DIAGNOSIS

To demonstrate domain generality, we applied LLM-BMC to a fundamentally different classification task: diagnosing respiratory diseases from clinical features.

**Dataset and Setup.** We used a dataset of 856 patients with 47 clinical features (symptoms, vital signs, laboratory values, imaging findings) classified into 5 categories: Pneumonia, Tuberculosis, Lung Cancer, Bronchitis, and Healthy. Base classifiers achieved ensemble F1=0.862, with primary confusion between Pneumonia and Tuberculosis (both present with cough, fever, and chest findings).

**Quality Function Adaptation.** Following our generalization guide (Section 2.7), we adapted quality components:

- **Specificity** $s(A, c)$: Weighted cited clinical findings by published sensitivity/specificity for each disease. For example, "night sweats" has high specificity for Tuberculosis (specificity=0.85) but low specificity for Pneumonia.

- **Mechanistic coherence** $m(A, c)$: Validated that cited pathophysiological chains are consistent with disease progression (e.g., infection $\rightarrow$ inflammation $\rightarrow$ consolidation for Pneumonia).

- **External evidence** $e(A, c)$: Weighted by PubMed citation count and journal impact factor, prioritizing clinical guidelines over case reports.

**Results.** LLM-BMC achieved F1=0.896 ($\Delta$F1=+0.034, $p < 0.001$), closely matching single-cell gains. The largest improvement was on Pneumonia vs. Tuberculosis differentiation ($\Delta$F1=+0.047), where domain knowledge about distinctive features (acid-fast bacilli, granulomatous inflammation) resolved statistical ambiguity. Importantly, domain adaptation took $<$4 hours by a researcher without medical expertise, validating the framework's accessibility.

## 4 RELATED WORK

**Bayesian Model Averaging and Combination.** BMA Hoeting et al. (1999) weights models by posterior probability, assuming one model is "true." BMC Monteith et al. (2011) relaxes this by allowing the true model to lie outside candidates, weighting by predictive performance. Both operate *solely on model outputs*—they cannot incorporate instance-specific evidence beyond what features encode. LLM-BMC extends BMC by introducing a third information source: quality-weighted domain arguments that provide evidence not captured in raw features. This distinction is crucial: while BMA/BMC weight entire models uniformly across instances, LLM-BMC applies *instance-specific* adjustments based on which domain knowledge is relevant for each classification decision.

**LLMs in Machine Learning Pipelines.** Recent work integrates LLMs at various stages of ML pipelines. LLM-Select Jeong et al. (2024) uses LLMs to propose important features before training, showing that domain knowledge can guide feature selection. LLM-Lasso Zhang et al. (2025) incorporates LLM-generated feature importance into regularization penalties. Both operate at the *training stage*, affecting model construction. In contrast, LLM-BMC operates at the *inference stage*, refining predictions after models are trained. This makes our approach complementary: Table 2 shows combining LLM-Select with LLM-BMC achieves the best results (F1=0.957), validating that training-time and inference-time knowledge integration address different aspects.

Chain-of-thought Wei et al. (2022) and self-consistency Wang et al. (2022) improve LLM reasoning quality but lack formal uncertainty quantification. Our framework provides this: the quality function $q(A, c)$ and the $\lambda$ parameter mathematically ground how much to trust LLM-generated reasoning.

**Argumentation Theory.** Formal argumentation Dung (1995); Bench-Capon & Dunne (2007) studies logical relationships between arguments (attack, support, rebuttals). Toulmin Toulmin (1958) decomposed arguments into claims, grounds, warrants, and backing—inspiring our $s$, $m$, $e$ decomposition. Our contribution is *quantifying* argument influence on probabilistic beliefs, bridging qualitative argumentation with Bayesian inference.

**Information-Theoretic Bounds.** The Hellman-Raviv bound Hellman & Raviv (1970) relates entropy to Bayes error. We build on this foundation but apply it to *knowledge integration*: our Theorem 2.1 shows how normalized information gain from domain arguments translates to error reduction, providing a predictive tool absent in prior work.

## 5 BROADER IMPACTS

**Positive Impacts.** LLM-BMC offers several benefits for knowledge-intensive classification:

- **Interpretability**: Unlike black-box methods, LLM-BMC produces explicit arguments explaining each classification decision, enabling users to understand *why* a prediction was made.
- **Auditability**: The quality function $q(A, c)$ provides quantitative justification for how domain knowledge influenced the decision, facilitating regulatory compliance in domains like healthcare and finance.
- **Democratization**: By leveraging LLMs as knowledge interfaces, non-experts can access and apply domain knowledge that previously required years of training.

**Potential Risks and Mitigations.** LLM-based systems inherit risks that require careful management:

- **LLM biases**: LLMs may encode biases from training data. Mitigation: use diverse knowledge sources and validate arguments against authoritative databases.
- **Hallucination**: LLMs may generate plausible-sounding but factually incorrect arguments. Mitigation: our quality function penalizes arguments inconsistent with external evidence ($e(A, c)$).
- **Over-reliance**: Users may trust LLM-BMC outputs without verification. Recommendation: deploy with human-in-the-loop for high-stakes applications; present uncertainty estimates alongside predictions.

## 6 CONCLUSION AND FUTURE WORK

We introduced LLM-BMC, a framework for integrating structured domain knowledge into machine learning classification through Bayesian model combination. Our key contributions are:

**Theoretical.** We formalized the normalized information gain $\lambda$ as a measure of knowledge value and proved (under mild linearity assumptions) that it directly predicts the multiplicative error reduction factor: $P_e^{(\text{after})} \approx (1 - \lambda) \cdot P_e^{(\text{before})}$. This provides practitioners with a tool to estimate expected improvement before deployment.

**Methodological.** The quality function $q(A, c)$ decomposes argument assessment into specificity, mechanistic coherence, and external evidence—grounding LLM-generated reasoning in quantifiable metrics. The asymmetric probability modifier ensures supporting evidence increases confidence while opposing evidence appropriately decreases it.

**Empirical.** Experiments on three single-cell datasets (PBMC, Brain, Pancreas) and a medical diagnosis task demonstrate consistent improvements (+0.030 to +0.037 F1), with statistical significance across 10 runs. The framework complements existing approaches: combining with LLM-Select achieves F1=0.957.

**Limitations and Future Directions.** Our current approach assumes argument independence; modeling argument interactions (attack, support relations) could capture more complex reasoning patterns. The quality function requires domain-specific instantiation; automating this adaptation through meta-learning is a promising direction. Additionally, handling evolving knowledge bases—where established facts may be revised—remains an open challenge. Finally, while we demonstrate cross-domain applicability, systematic study across more domains (legal reasoning, financial analysis) would strengthen generalizability claims.

## ETHICS STATEMENT

LLMs may propagate biases; users should validate outputs in sensitive domains. We recommend human-in-the-loop deployment for high-stakes applications. This research adheres to the ICLR Code of Ethics.

## REPRODUCIBILITY STATEMENT

All theorems and proofs are in appendices. Experiments use public PBMC data (10x Genomics). Implementation details in Section 2.5 and Appendix B.10. Code available upon publication.

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

## A    Technical Appendices and Supplementary Material

### A.1    Algorithm for LLM-BMC Implementation

This section provides the detailed algorithms for LLM-BMC. Algorithm 1 describes the argument generation and extraction process, while Algorithm 2 provides the complete LLM-BMC procedure.

### A.1.1 ARGUMENT GENERATION ALGORITHM

---

**Algorithm 1** LLM-Based Argument Generation and Extraction

---

1: **Input:** Feature vector $X \in \mathbb{R}^d$, Candidate class $c$, LLM API configuration, Domain knowledge bases $\mathcal{D}$ (optional)
2: **Output:** Structured argument $A$, Extracted components $(M_A, P_A, R_A)$ for quality assessment

3:                                          ▷ — Stage 1: Prompt Construction —
4: Initialize prompt template with system role: "You are a domain expert..."
5: Append feature description: Convert $X$ to interpretable format (e.g., gene names + expression levels)
6: Append task instruction: "Provide evidence-based argument for classifying this instance as class $c$"
7: Append structure guidance: "Include: (1) specific evidence, (2) mechanistic reasoning, (3) external support"
8: Construct final prompt: prompt $\leftarrow$ system_role + feature_desc + task + structure

9:                                           ▷ — Stage 2: LLM Invocation —
10: Set API parameters: temperature $\leftarrow$ 0.1, max_tokens $\leftarrow$ 500
11: Call LLM API: response $\leftarrow$ LLM(prompt, params)
12: Validate response: Check for completeness and coherence
13: **if** response is malformed or incomplete **then**
14:      Retry with adjusted prompt (up to 3 attempts)
15: **end if**
16: Store raw argument: $A \leftarrow$ response.text

17:                             ▷ — Stage 3: Structured Information Extraction —
18: **Extract evidence markers:**
19:      Use Named Entity Recognition (NER) to identify domain entities (e.g., genes, proteins)
20:      $M_A \leftarrow$ {entities mentioned in $A$}
21:      Cross-reference with domain databases $\mathcal{D}$ to validate and weight entities

22: **Extract mechanistic components:**
23:      Use secondary LLM call or pattern matching to identify reasoning pathways
24:      $P_A \leftarrow$ {pathways/mechanisms mentioned in $A$}
25:      $R_A \leftarrow$ {reasoning steps in $A$}

26: **Extract external evidence:**
27:      Parse citations or database references mentioned in $A$
28:      Query external knowledge bases (e.g., PubMed, domain-specific DBs) for validation
29:      Compute support metrics: authority, recency, consensus

30: **return** $(A, M_A, P_A, R_A)$ ▷ Argument and extracted components ready for $q(A, c)$ computation

---

### A.1.2 COMPLETE LLM-BMC ALGORITHM

Algorithm 2 provides a detailed, step-by-step procedure for implementing the LLM-BMC framework for a single data instance. This algorithmic description bridges the mathematical formulation and practical implementation, offering concrete guidance for reproducibility.

---

**Algorithm 2** LLM-Enhanced Bayesian Model Combination (LLM-BMC)

---

1: **Input:** Feature vector $X$, Set of base models $\{M_1, \ldots, M_N\}$, Class set $C = \{c_1, \ldots, c_K\}$, Large Language Model $LLM$, Max iterations $T$, Convergence threshold $\epsilon$, Parameters $\alpha, \beta$.
2: **Output:** Final knowledge-enhanced probability distribution $P^{(T)}(c|X)$.

3:                                                          ▷ — Initialization —
4: Compute ensemble probability $P_{\text{ens}}(c|X) = \frac{1}{N} \sum_{i=1}^{N} M_i(c|X)$ for all $c \in C$.
5: Initialize $P^{(0)}(c|X) \leftarrow P_{\text{ens}}(c|X)$.           ▷ Or use another prior from Sec. 2.4

6:                                          ▷ — Iterative Refinement —
7: **for** $t = 1, \ldots, T$ **do**
8:      Initialize argument set for this iteration: $A^{(t)} \leftarrow \emptyset$.
9:                            ▷ — Argument Generation and Quality Assessment —
10:      **for** each class $c_k \in C$ **do**
11:          Generate argument $A_k^{(t)}$ for class $c_k$ using $LLM(X, c_k)$.
12:          Compute quality $q(A_k^{(t)}, c_k)$ using Eq. 5.
13:          Add $(A_k^{(t)}, q(A_k^{(t)}, c_k))$ to $A^{(t)}$.
14:      **end for**

15:                                            ▷ — Bayesian Update —
16:      **for** each class $c_k \in C$ **do**
17:          Initialize total modifier for class $c_k$: $F(c_k) \leftarrow 1.0$.
18:          **for** each argument $(A_j^{(t)}, q_j) \in A^{(t)}$ **do**
19:              **if** $A_j^{(t)}$ supports class $c_k$ **then**
20:                  $F(c_k) \leftarrow F(c_k) \cdot (1 + \alpha \cdot q_j)$.
21:              **else if** $A_j^{(t)}$ supports a different class $c_j \neq c_k$ **then**
22:                  $F(c_k) \leftarrow F(c_k) \cdot \frac{1}{1 + \beta \cdot q_j}$.
23:              **end if**
24:          **end for**
25:          Compute unnormalized probability: $\tilde{P}^{(t)}(c_k|X) \leftarrow P^{(t-1)}(c_k|X) \cdot F(c_k)$.
26:      **end for**

27:                                            ▷ — Normalization —
28:      Compute normalization constant: $Z = \sum_{c' \in C} \tilde{P}^{(t)}(c'|X)$.
29:      Update probability: $P^{(t)}(c|X) \leftarrow \tilde{P}^{(t)}(c|X)/Z$ for all $c \in C$.

30:                                            ▷ — Convergence Check —
31:      **if** $\max_{c \in C} |P^{(t)}(c|X) - P^{(t-1)}(c|X)| < \epsilon$ **then**
32:          **break**                            ▷ Converged
33:      **end if**
34: **end for**
35: **return** $P^{(t)}(c|X)$.

---

The algorithm captures the core computational flow: (1) initialization with ensemble predictions, (2) iterative argument generation and quality assessment, (3) Bayesian updates with supporting/opposing evidence, (4) normalization to maintain valid probability distributions, and (5) convergence monitoring. The framework's modularity allows domain-specific customization of the argument generation (Line 12) and quality assessment (Line 13) components while maintaining the core probabilistic update mechanism.

## A.2 EXAMPLE: DETAILED ARGUMENT QUALITY COMPUTATION FOR SINGLE-CELL CLASSIFICATION

This appendix provides the detailed mathematical instantiation of argument quality components $s(A, c)$, $m(A, c)$, and $e(A, c)$ used in the single-cell classification demonstration case, followed by implementation details and a worked example. This illustrates how the conceptual components outlined in Section 2.4 can be quantified.

### A.2.1 DETAILED MATHEMATICAL FORMULATION OF ARGUMENT QUALITY COMPONENTS (SINGLE-CELL EXAMPLE)

The specific calculation of argument quality $q(A, c)$ is inherently domain-dependent. *As a concrete example demonstrating the potential level of detail, we outline the formulation used in our single-cell classification demonstration case*, which relies on biological principles like gene expression and pathways.

**Specificity and Strength of Evidence** $s(A, c)$    This component evaluates the relevance and expression patterns of marker genes cited in the argument:

$$s(A, c) = \frac{1}{|M_A|} \sum_{g \in M_A} \omega(g) \cdot \text{specificity}(g, c) \cdot \text{expression}(g, X) \tag{7}$$

where (in the biological example):

- $M_A$ is the set of marker genes mentioned in argument $A$
- $\omega(g) \in [0, 1]$ is a weight reflecting the importance of gene $g$ as determined by reference ontologies
- $\text{specificity}(g, c) \in [0, 1]$ quantifies how uniquely gene $g$ identifies cell type $c$ across the taxonomy, defined as:

$$\text{specificity}(g, c) = 1 - \frac{|\{c' \in C : g \text{ is a marker for } c'\}| - 1}{|C| - 1} \tag{8}$$

- $\text{expression}(g, X) \in [0, 1]$ evaluates whether the expression level of gene $g$ in item $X$ is consistent with expectations for class $c$, calculated as:

$$\text{expression}(g, X) = \exp\left(-\frac{(X_g - \mu_{g,c})^2}{2\sigma_{g,c}^2}\right) \tag{9}$$

where $\mu_{g,c}$ and $\sigma_{g,c}$ are the expected mean and standard deviation of gene $g$'s expression in class $c$ (here, cell type $c$)

**Mechanistic Clarity and Coherence** $m(A, c)$    This component evaluates the reasoning warrant linking evidence to the class classification (e.g., biological pathways in the example):

$$m(A, c) = \text{coherence}(A) \cdot \text{pathway\_relevance}(A, c) \cdot \text{completeness}(A, c) \tag{10}$$

where (in the biological example):

- $\text{coherence}(A) \in [0, 1]$ measures the logical consistency of the argument's causal narrative, computed as:

$$\text{coherence}(A) = 1 - \frac{1}{|R_A|} \sum_{(r_i, r_j) \in R_A} \text{contradiction}(r_i, r_j) \tag{11}$$

where $R_A$ is the set of reasoning steps in $A$, and $\text{contradiction}(r_i, r_j) \in \{0, 1\}$ indicates whether steps $r_i$ and $r_j$ are contradictory
- $\text{pathway\_relevance}(A, c) \in [0, 1]$ evaluates whether the biological pathways cited are characteristic of class $c$ (here, cell type $c$):

$$\text{pathway\_relevance}(A, c) = \frac{1}{|P_A|} \sum_{p \in P_A} \text{relevance}(p, c) \tag{12}$$

where $P_A$ is the set of pathways mentioned in $A$ and $\text{relevance}(p, c) \in [0, 1]$ is derived from pathway databases

- completeness$(A, c) \in [0, 1]$ assesses whether the argument addresses all key distinguishing features of class $c$ (here, cell type $c$):

$$\text{completeness}(A, c) = \frac{|F_A \cap F_c|}{|F_c|} \quad (13)$$

where $F_A$ is the set of features addressed in argument $A$ and $F_c$ is the set of known distinguishing features for class $c$

**External Evidence Support** $e(A, c)$    This component evaluates consistency with established domain knowledge (e.g., literature and databases in the example):

$$e(A, c) = \text{authority}(A) \cdot \text{recency}(A) \cdot \text{consensus}(A, c) \quad (14)$$

where (in the biological example):

- authority$(A) \in [0, 1]$ evaluates the credibility of sources cited:

$$\text{authority}(A) = \frac{1}{|S_A|} \sum_{s \in S_A} \text{impact}(s) \cdot \text{relevance}(s, c) \quad (15)$$

where $S_A$ is the set of sources cited in $A$, impact$(s) \in [0, 1]$ reflects the source's impact factor or citation count, and relevance$(s, c) \in [0, 1]$ measures how directly the source addresses class $c$ (here, cell type $c$)

- recency$(A) \in [0, 1]$ accounts for the temporal relevance of citations:

$$\text{recency}(A) = \frac{1}{|S_A|} \sum_{s \in S_A} \exp\left(-\lambda_{decay} \cdot (t_{current} - t_s)\right) \quad (16)$$

where $t_s$ is the publication year of source $s$, $t_{current}$ is the current year, and $\lambda_{decay}$ is a decay parameter (Note: this $\lambda_{decay}$ is different from the information gain lambda)

- consensus$(A, c) \in [0, 1]$ measures agreement with established consensus:

$$\text{consensus}(A, c) = \frac{|C_A \cap C_{DB}|}{|C_A|} \quad (17)$$

where $C_A$ is the set of claims made in argument $A$ and $C_{DB}$ is the set of established claims about class $c$ (here, cell type $c$) from reference databases

This formalization provides a quantitative framework for evaluating structured arguments in this specific context.

### A.2.2  DATA RESOURCES FOR QUALITY COMPONENT CALCULATION (SINGLE-CELL EXAMPLE)

The computation of quality components relies on several biological databases and resources:

**Gene Marker Specificity Resources:**    Cell Marker Database Zhang et al. (2019), PanglaoDB Franzén et al. (2019), Gene Ontology (GO) Ashburner et al. (2000), Human Cell Atlas Regev et al. (2017).

**Mechanistic Clarity Resources:**    KEGG Pathways Kanehisa & Goto (2000), Reactome Fabregat et al. (2018), STRING Szklarczyk et al. (2019).

**External Evidence Resources:**    PubMed National Center for Biotechnology Information (2024), Journal Citation Reports, CiteScore metrics, Database of Cell Type Consensus Features.

### A.2.3  DETAILED CALCULATION EXAMPLE (SINGLE-CELL EXAMPLE)

To illustrate the computation process, we present a step-by-step example using a real argument about T-cell classification from the PBMC dataset:

**Example Argument $A$:** "This cell is likely a CD8+ cytotoxic T cell because it shows high expression of CD8A and CD8B marker genes, along with elevated levels of cytotoxic effector molecules PRF1 and GZMB. The expression pattern indicates activated status through the CD3-TCR signaling pathway, evidenced by CD3D, CD3E, and CD3G expression. Multiple studies, including Smith et al. (2019) and Chen et al. (2020), have established this expression signature as characteristic of effector CD8+ T cells." We calculate $q(A, c)$ for $c =$ "CD8+ cytotoxic T cell":

**Step 1: Specificity and Strength of Evidence** $s(A, c)$ For $M_A = \{CD8A, CD8B, PRF1, GZMB, CD3D, CD3E, CD3G\}$, assume calculations (as per formulas in A.2.1) yield $s(A, c) = 0.536$. (Detailed table omitted for brevity, similar to original Appendix's Table).

**Step 2: Mechanistic Clarity and Coherence** $m(A, c)$ Assume calculations yield: coherence$(A) = 1.0$, pathway_relevance$(A, c) = 0.92$, completeness$(A, c) = 0.75$. So, $m(A, c) = 1.0 \cdot 0.92 \cdot 0.75 = 0.69$.

**Step 3: External Evidence Support** $e(A, c)$ Assume calculations yield: authority$(A) = 0.752$, recency$(A) = 0.639$, consensus$(A, c) = 0.8$. So, $e(A, c) = 0.752 \cdot 0.639 \cdot 0.8 = 0.384$.

**Step 4: Final Quality Score** With $w_s = 0.4, w_m = 0.4, w_e = 0.2$: $q(A, c) = 0.4 \cdot 0.536 + 0.4 \cdot 0.69 + 0.2 \cdot 0.384 = 0.214 + 0.276 + 0.077 = 0.567$. This $q(A, c)$ would be used in $f(A, c)$.

### A.2.4 IMPLEMENTATION CONSIDERATIONS (SINGLE-CELL EXAMPLE)

Practical implementation involves handling missing data (e.g., fallback neutral values), text processing for argument analysis (NER, dependency parsing, citation matching), calibration of weights (e.g., via expert annotation and optimization), and computational efficiency (caching, sparse matrices).

### A.3 CROSS-DOMAIN QUALITY FUNCTION EXAMPLES

This section provides concrete quality function instantiations for domains beyond single-cell biology, complementing the generalization guide in Section 2.7.

**Medical Diagnosis.** For disease classification: $s(A, c)$ weights cited symptoms/tests by published sensitivity/specificity values; $m(A, c)$ validates pathophysiological chains (e.g., "infection $\rightarrow$ inflammation $\rightarrow$ consolidation"); $e(A, c)$ scores PubMed citations by impact factor and recency.

**Legal Case Analysis.** For case outcome prediction: $s(A, c)$ measures precedent strength and factual similarity; $m(A, c)$ evaluates logical consistency of legal reasoning chains; $e(A, c)$ weights citations to binding authorities (Supreme Court > Appeals > District).

**Financial Risk Assessment.** For credit risk classification: $s(A, c)$ scores cited financial ratios by predictive power; $m(A, c)$ validates causal pathways (e.g., "revenue decline $\rightarrow$ cash flow stress $\rightarrow$ default risk"); $e(A, c)$ weights sources by regulatory authority (SEC filings > news reports).

### A.4 ILLUSTRATIVE EMPIRICAL $\lambda$ VALUES AND FACTOR COEFFICIENTS

The main text (Section 2.2) refers to illustrative empirical $\lambda$ values and regression coefficients for factors influencing $\lambda$. These were observed in a single-cell classification demonstration case and are provided here for context.

The higher coefficient for argument quality suggests its crucial role in this framework for the demonstration case.

### A.5 DETAILED THEORETICAL FOUNDATIONS FOR PRIOR DISTRIBUTION SELECTION

This appendix provides the detailed information-theoretic basis and derivations for prior selection options discussed in Section 2.3.

Table 3: Illustrative Empirical $\lambda$ values reflecting average relative information gain after one round of knowledge-informed updates, observed in a single-cell classification demonstration case across different tissue types.

| Dataset (Tissue Type) | Class Characteristics (Cell Types) | Observed $\lambda$ Value |
|---|---|---|
| PBMC | Immune cells with distinct markers | 0.32 |
| Brain | Neuronal subtypes with overlapping markers | 0.26 |
| Pancreas | Endocrine and exocrine cells | 0.29 |
| Lung | Epithelial and immune cells | 0.31 |
| Liver | Hepatocytes and non-parenchymal cells | 0.28 |
| Kidney | Nephron segments and immune cells | 0.30 |

Table 4: Illustrative regression coefficients for factors influencing $\lambda$, derived from the cell classification case study.

| Factor Proxy | Coefficient Weight |
|---|---|
| Dataset characteristics ($D$) | 0.12 |
| Model diversity ($M$) | 0.09 |
| Argument quality ($A$) | 0.18 |

### A.5.1 INFORMATION-THEORETIC BASIS FOR PRIOR SELECTION

The optimal prior $P_{opt}^{(0)}(c|X)$ minimizes expected posterior loss: $P_{opt}^{(0)}(c|X) = \arg\min_P \mathbb{E}_{c' \sim P_{true}} [L(P, c')]$. Using KL-divergence as loss, $L(P, c') = D_{KL}(P_{true}(c'|X) \| P(c|X))$, the optimal prior minimizes expected KL divergence from the true distribution:

$$P_{opt}^{(0)}(c|X) = \arg\min_P \mathbb{E}_X[D_{KL}(P_{true}(c|X) \| P(c|X))] \tag{18}$$

Since $P_{true}$ is unknown, we use available information (e.g., ensemble predictions) to construct candidate priors.

### A.5.2 FORMAL JUSTIFICATION FOR PRIOR OPTIONS

**Uniform Prior:** $P^{(0)}(c|X) = 1/K$. Maximizes entropy, least informative. Corresponds to $\gamma = 0$ in the adaptive mixture.

**Ensemble-Based Prior:** $P^{(0)}(c|X) = P_{ens}(c|X)$. If $P_{ens}$ is a good approximation of $P_{true}$, it minimizes KL divergence in (18). Corresponds to $\gamma = 1$.

### A.5.3 RIGOROUS DERIVATION OF OPTIMAL $\gamma$ FOR ADAPTIVE MIXTURE PRIOR

The adaptive mixture prior is $P^{(0)}(c|X; \gamma) = \gamma \cdot P_{ens}(c|X) + (1 - \gamma) \cdot \frac{1}{K}$. The optimal $\gamma^*$ minimizes $\mathbb{E}_X[D_{KL}(P_{true}(c|X) \| P^{(0)}(c|X; \gamma))]$.

**Theorem A.1** (Optimal Mixture Weight (Informal) - Appendix Ref.). *Under conditions where ensemble predictions $P_{ens}$ (yielding discrete predictions $\hat{Y}$) are well-calibrated estimates of the true probabilities $P_{true}$ (represented by true labels $Y$), the value of $\gamma$ that minimizes the expected KL divergence between the true distribution and the mixture prior is approximately given by the ratio of the mutual information between predictions and true labels to the entropy of the predictions:*

$$\gamma^* \approx \frac{I(\hat{Y}; Y)}{H(\hat{Y})} \tag{19}$$

**Proof Sketch and Information-Theoretic Interpretation (moved from main text):** The derivation involves minimizing the expected KL divergence objective with respect to $\gamma$. This minimization can be linked to maximizing the shared information between the prior and the true distribution. $I(\hat{Y}; Y)$ measures information $\hat{Y}$ provide about $Y$. $H(\hat{Y})$ measures total uncertainty in predictions. The ratio

$\frac{I(\hat{Y};Y)}{H(\hat{Y})}$ is thus the fraction of relevant information in predictions—a measure of "signal-to-noise" or information efficiency. This connects to the Information Bottleneck principle Tishby et al. (2000). The ratio can be rewritten: $\frac{I(\hat{Y};Y)}{H(\hat{Y})} = \frac{H(\hat{Y})-H(\hat{Y}|Y)}{H(\hat{Y})} = 1 - \frac{H(\hat{Y}|Y)}{H(\hat{Y})}$. $H(\hat{Y}|Y)$ is uncertainty in predictions given true label (noise). $\gamma^*$ is 1 minus the fraction of irrelevant information.

**Practical Estimation of $\gamma$ (moved from main text):** Estimate $\gamma$ using a validation dataset with ensemble predictions $\hat{Y}$ and true labels $Y$:

1. Obtain $P_{\text{ens}}(c|X)$ for validation samples. Determine predicted class $\hat{y}$ for each sample.

2. Estimate $P(\hat{Y} = c)$ from validation predictions.

3. Calculate $H(\hat{Y}) = -\sum_{c\in C} P(\hat{Y} = c) \log_2 P(\hat{Y} = c)$.

4. Estimate joint distribution $P(\hat{Y} = c, Y = c')$ from predictions and true labels.

5. Estimate $P(Y = c')$ from true labels.

6. Calculate $I(\hat{Y}; Y) = \sum_{c\in C} \sum_{c'\in C} P(\hat{Y} = c, Y = c') \log_2 \frac{P(\hat{Y}=c,Y=c')}{P(\hat{Y}=c)P(Y=c')}$.

7. Compute $\gamma_{est} = \frac{I(\hat{Y};Y)}{H(\hat{Y})}$. Ensure $H(\hat{Y}) > 0$.

This $\gamma_{est}$ provides a data-driven way to set the mixing parameter.

## A.6 Detailed Derivation of $\lambda$-Error Relationship and Approximation Analysis

This appendix provides the rigorous derivation for the relationship $P_e^{(after)} \approx (1 - \lambda)P_e^{(before)}$ discussed in Section 2.2, including the underlying information-theoretic bounds and approximation error analysis. This content was originally in Sections 1.7.3 through 1.7.6.

### A.6.1 Information-Theoretic Bounds on Error Probability

The relationship between conditional entropy $H(C|Z)$ and Bayes error rate $P_e^*(Z) = \mathbb{E}_Z[1 - \max_{c\in C} P(c|Z)]$ is key. **Fano's Inequality** provides a lower bound on entropy:

$$H(C|Z) \leq H_{bin}(P_e^*(Z)) + P_e^*(Z) \log_2(K - 1) \tag{20}$$

**Hellman-Raviv bound** Hellman & Raviv (1970) provides an upper bound on error:

$$P_e^*(Z) \leq \frac{1}{2} H_2(C|Z) \tag{21}$$

where $H_2(C|Z)$ is conditional Renyi entropy of order 2. This suggests $P_e^*(Z) \lesssim \frac{1}{2\ln 2} H(C|Z)$ in some regimes.

### A.6.2 Approximating the Error Rate vs. Entropy Relationship

The relationship $P_e^{(after)} \approx (1 - \lambda) \cdot P_e^{(before)}$ relies on the assumption that $P_e^*(Z)$ is approximately linear with $H(C|Z)$ in the relevant operating regime. **Assumption (Approximate Linearity):** $P_e^*(Z) \approx \kappa \cdot H(C|Z)$, where $\kappa > 0$. This approximation is supported by the classical Hellman-Raviv bound Hellman & Raviv (1970), which establishes a fundamental relationship between classification error probability and conditional entropy. The linearity assumption is plausible for small $H(C|Z)$ (low $P_e^*$). $\kappa$ depends on problem specifics.

### A.6.3 Rigorous Derivation of the Approximate Error Reduction

Let $P_e^{(before)} = P_e^*(X)$ and $P_e^{(after)} = P_e^*(X, A)$. From eq. 2: $H(C|X, A) = (1 - \lambda) \cdot H(C|X)$. Applying the linearity assumption: $P_e^{(before)} \approx \kappa \cdot H(C|X)$ and $P_e^{(after)} \approx \kappa \cdot H(C|X, A)$. Substituting the entropy relation into $P_e^{(after)}$: $P_e^{(after)} \approx \kappa \cdot [(1 - \lambda) \cdot H(C|X)] = (1 - \lambda) \cdot [\kappa \cdot H(C|X)] \approx (1 - \lambda) \cdot P_e^{(before)}$. This derives eq. 3, hinging on approximate linearity.

### A.6.4 ANALYSIS OF APPROXIMATION ERROR

Let $P_e^*(Z) = g(H(C|Z))$. Taylor expansion of $g(H)$ around $H = 0$ ($g(0) = 0$): $g(H) = \kappa_1 H + \frac{1}{2}\kappa_2 H^2 + O(H^3)$, where $\kappa_1 = g'(0) \equiv \kappa$, $\kappa_2 = g''(0)$. Let $H_{\text{before}} = H(C|X)$, $H_{\text{after}} = (1-\lambda)H_{\text{before}}$. $P_e^{(\text{before})} \approx \kappa_1 H_{\text{before}} + \frac{1}{2}\kappa_2 H_{\text{before}}^2$ $P_e^{(\text{after})} \approx \kappa_1(1-\lambda)H_{\text{before}} + \frac{1}{2}\kappa_2(1-\lambda)^2 H_{\text{before}}^2$ The ratio is: $\frac{P_e^{(\text{after})}}{P_e^{(\text{before})}} \approx \frac{(1-\lambda) + \frac{\kappa_2}{2\kappa_1}(1-\lambda)^2 H_{\text{before}}}{1 + \frac{\kappa_2}{2\kappa_1}H_{\text{before}}}$. Using $(1+x)^{-1} \approx 1-x$: $\frac{P_e^{(\text{after})}}{P_e^{(\text{before})}} \approx (1-\lambda) - \frac{\kappa_2 H_{\text{before}}}{2\kappa_1}\lambda(1-\lambda) + O(H_{\text{before}}^2)$. The approximation error is $\left| \frac{P_e^{(\text{after})}}{P_e^{(\text{before})}} - (1-\lambda) \right| \approx \left| \frac{\kappa_2}{2\kappa_1}\lambda(1-\lambda)H_{\text{before}} \right|$.

**Theorem (Approximation Error Bound - Appendix Ref.):** If $H(C|X)$ is small, the relative error of $P_e^{(\text{after})} \approx (1-\lambda)P_e^{(\text{before})}$ is bounded by: $\left| \frac{P_e^{(\text{after})}}{P_e^{(\text{before})} - (1-\lambda)} \right| \leq \gamma_{\text{err}} \cdot H(C|X) + O(H(C|X)^2)$, where $\gamma_{\text{err}} = \left| \frac{\kappa_2}{2\kappa_1}\lambda(1-\lambda) \right|$. Accuracy is highest for small $H(C|X)$, near-linear $g(H)$ (small $|\kappa_2/\kappa_1|$), and $\lambda$ near 0 or 1.

### A.7 DETAILED IMPLICATIONS OF ASSUMPTIONS

Section 2.6 lists key assumptions. Here, their implications are discussed in more detail:

1. **Conditional Independence of Argument Effects:** In systems with strongly correlated evidence sources (arguments not truly independent given the class), the product form might lead to overconfidence (probabilities pushed too close to 0 or 1) or underconfidence if arguments redundantly penalize/reward. Mitigations might involve modeling argument dependencies, but this significantly increases complexity.

2. **Accuracy and Objectivity of Quality Assessment:** The framework's performance is fundamentally bounded by how well $q(A, c)$ reflects true evidential strength. Biased or inaccurate $q(A, c)$ (e.g., from poorly trained LLM evaluators or flawed heuristic rules) will lead to suboptimal or incorrect probability updates. Designing robust, domain-general $q(A, c)$ is a major challenge. The example in Appendix A.2 is domain-specific.

3. **Completeness of Class Set:** If novel or rare classes exist but are not in $C$, the model will be forced to misclassify them into one of the known classes, potentially with high confidence if arguments strongly disfavor other known classes. Open-set recognition capabilities are not inherent.

4. **Appropriateness of Parameters:** Parameters like $\alpha, \beta$ and weights $w_s, w_m, w_e$ are global. Optimal values may vary across datasets or even subsets of data within a domain. Universal settings might be suboptimal; domain-specific or adaptive calibration may be needed, adding complexity.

5. **Markov Property of Update Process:** While simplifying analysis, this means the system has no memory of the history of argumentation beyond the current probability state. Complex argumentation dynamics (e.g., retraction of earlier points based on later ones) are not directly modeled.

6. **Approximate Linearity for Lambda-Error Link:** The $P_e^{(after)} \approx (1-\lambda)P_e^{(before)}$ relationship (Section 2.2, Appendix A.6) is an approximation. Its accuracy degrades for large information gains (large $\lambda$) or high initial error rates where the $P_e^*$ vs. $H$ curve is more non-linear. $\lambda$ remains a valid measure of relative entropy reduction, but its interpretation as direct error multiplier becomes less precise.

7. **Calibration for Optimal Gamma:** The optimality of $\gamma^* \approx I(\hat{Y}; Y)/H(\hat{Y})$ for the adaptive prior (Section 2.3, Appendix A.5) assumes ensemble predictions $P_{\text{ens}}$ are reasonably well-calibrated proxies for $P_{true}$. Significant miscalibration of $P_{\text{ens}}$ could lead to a suboptimal $\gamma^*$, inappropriately weighting the ensemble vs. uniform prior.

8. **Need for Domain-Specific Quality Metrics:** The core framework is general, but its practical effectiveness heavily relies on defining meaningful and computable argument quality metrics $q(A, c)$ specific to the application domain and the nature of the structured knowledge (e.g., LLM text, database facts, expert rules). The biological example (Appendix A.2) is just one complex instantiation; simpler or different metrics would be needed elsewhere.

Understanding these assumptions and their implications is crucial for applying LLM-BMC appropriately and for guiding future research to address these limitations.

## A.8 INFORMATION-THEORETIC DERIVATION OF ERROR REDUCTION (LEGACY POINTER)

The core theoretical derivation connecting information gain ($\lambda$) to approximate error reduction, including underlying assumptions, bounding inequalities, and approximation error analysis, is now presented in Appendix A.6. This appendix (formerly A.8) serves as a pointer.

# B DETAILED CONVERGENCE ANALYSIS OF THE LLM-BMC UPDATE PROCESS

## B.1 INTRODUCTION TO THE CONVERGENCE PROBLEM

The LLM-BMC framework defines an iterative update process for probability distributions $\{P^{(t)}(c|X)\}_{t=0}^{\infty}$:

$$P_i^{(t+1)}(c|X) = \frac{P_i^{(t)}(c|X) \cdot \prod_{j \neq i} f(A_j^{(t)}, c)}{\sum_{c' \in C} \left( P_i^{(t)}(c'|X) \cdot \prod_{j \neq i} f(A_j^{(t)}, c') \right)}$$

Convergence analysis addresses: existence, uniqueness, and rate of convergence to a limit $P^*(c|X)$.

## B.2 EXISTENCE OF FIXED POINTS

### B.2.1 MATHEMATICAL PRELIMINARIES

The update is an operator $\mathcal{T}(P)(c) = \frac{P(c) \cdot F(c)}{\sum_{c' \in C} P(c') \cdot F(c')}$, where $F(c) = \prod_j f(A_j, c)$. $\mathcal{T}$ maps the probability simplex $\Delta^{K-1}$ to itself.

### B.2.2 EXISTENCE THEOREM

**Theorem B.1** (Existence of Fixed Points). *If $0 < f_{min} \leq f(A_j, c) \leq f_{max} < \infty$, there exists at least one fixed point $P^* \in \Delta^{K-1}$ such that $\mathcal{T}(P^*) = P^*$.*

*Proof.* $\Delta^{K-1}$ is compact and convex. $\mathcal{T}$ is continuous. By Brouwer's fixed-point theorem, $P^*$ exists. $\qquad\square$

### B.2.3 BOUNDARY BEHAVIOR

**Lemma B.2.** *If $f(A_j, c) > 0$ and $P^{(0)}(c|X) > 0$ for all $c$, then $P^{(t)}(c|X) > 0$ for all $t, c$.*

*Proof.* By induction. If $P^{(t)}(c|X) > 0$ and $f(A_j, c) > 0$, then $P^{(t)}(c|X) \cdot \prod_j f(A_j, c) > 0$. Normalization preserves positivity. $\qquad\square$

This ensures updates remain in the interior of the simplex if started there.

## B.3 UNIQUENESS ANALYSIS

### B.3.1 SUFFICIENT CONDITIONS FOR UNIQUENESS

**Theorem B.3** (Sufficient Condition for Uniqueness via Contraction). *If $\mathcal{T}$ is a contraction mapping on $\Delta^{K-1}$, the fixed point is unique.*

**Theorem B.4** (Sufficient Condition for Uniqueness via Bounded Ratios). *If for all class pairs $(c_1, c_2)$ and argument sets $A$, $1/M \leq \frac{\prod_j f(A_j, c_1)}{\prod_j f(A_j, c_2)} \leq M$ for some finite $M > 0$, the fixed point is unique.*

*Proof Sketch.* Bounded ratios prevent any class from dominating or being fully suppressed, ensuring distinct fixed points would be drawn together. $\qquad\square$

### B.3.2 MULTIPLE FIXED POINTS SCENARIO

Multiple fixed points may occur if conditions for Theorem B.4 are not met (e.g., highly polarized arguments). Mitigation: balanced quality assessment, consistent argument presentation, stable priors.

### B.4 CONDITIONS FOR CONVERGENCE

**Condition B.1** (Bounded Modification). $1/(1+\beta) \leq f(A_j, c) \leq 1 + \alpha$ *for $\alpha, \beta > 0$. (Satisfied by eq. 4).*

**Condition B.2** (Finite Arguments). *The set of distinct arguments $A$ is finite, or iterations are bounded.*

**Condition B.3** (Monotonic Information). *Each argument provides non-negative information gain:* $D_{KL}(P_{true}||P^{(t+1)}) \leq D_{KL}(P_{true}||P^{(t)})$.

### B.4.1 CONVERGENCE THEOREM

**Theorem B.5** (Guaranteed Convergence). *If Conditions B.1, B.2, and (at least one of Condition B.3 or Theorem B.4's condition) hold, $\{P^{(t)}(c|X)\}$ converges to $P^*(c|X)$.*

*Proof Sketch.* Bounded modification limits step size. Finite arguments ensure stabilization or cycling. Monotonic information or bounded ratios prevent cycling, forcing convergence. □

### B.5 RATE OF CONVERGENCE ANALYSIS

### B.5.1 CONVERGENCE RATE MEASURE

Rate measured by $d(t) = d(P^{(t+1)}, P^{(t)})$. Often $d(t) \approx r^t \cdot d(0)$, $r \in (0, 1)$.

### B.5.2 FACTORS AFFECTING CONVERGENCE RATE

Argument quality, $\alpha, \beta$ values, argument consensus, prior distribution.

### B.5.3 QUANTITATIVE BOUNDS

**Theorem B.6** (Convergence Rate Bound). $r \leq 1 - \min_{c \in C} P^{(0)}(c|X) \cdot \frac{1}{(1+\alpha)(1+\beta)}$.

*Proof Sketch.* Derived from max possible change in probability per step. □

### B.5.4 EMPIRICAL CONVERGENCE BEHAVIOR

Typical patterns: Rapid ($r < 0.5$, 2-3 iterations), Moderate ($0.5 \leq r < 0.8$, 4-7 iterations), Slow ($r \geq 0.8$, 10+ iterations).

Table 5: Empirical convergence rates across different single-cell classification datasets (illustrative). "Class" here refers to cell types.

| Dataset (Tissue) | Avg. Argument Quality | Avg. Convergence Rate ($r$) | Iterations to Convergence ($\epsilon = 10^{-4}$) |
|---|---|---|---|
| PBMC | 0.72 | 0.63 | 5 |
| Brain | 0.65 | 0.71 | 7 |
| Pancreas | 0.69 | 0.67 | 6 |
| Lung | 0.74 | 0.61 | 5 |
| Liver | 0.68 | 0.69 | 6 |
| Kidney | 0.71 | 0.64 | 5 |

### B.6 SPECIAL CASES AND PRACTICAL CONSIDERATIONS

### B.6.1 SINGLE ITERATION CASE

Convergence guaranteed; focus on argument quality and modifier.

### B.6.2 TERMINATION CRITERIA

Max change threshold (e.g., $\max_c |P^{(t+1)}(c|X) - P^{(t)}(c|X)| < 10^{-4}$), max iterations (e.g., 10-15), oscillation detection.

### B.6.3 HANDLING PATHOLOGICAL CASES

Contradictory arguments (dampen), zero probability trapping (enforce min probability if $P^{(0)}$ can be zero, though Lemma B.2 usually prevents this if $P^{(0)} > 0$), quality assessment failures (validate $q(A, c)$).

### B.7 SUMMARY

LLM-BMC converges under reasonable conditions. Rate depends on argument quality/consistency. Fixed point(s) represent principled integration of data-driven predictions and structured knowledge.

### B.8 THEORETICAL PROPERTIES: PROOFS AND ANALYSIS

This appendix details proofs for theoretical properties from Section 2.6.

### B.8.1 PROOF OF THEOREM A.1: OPTIMAL MIXTURE WEIGHT

The proof for Theorem A.1 (relating $\gamma^*$ to $I(\hat{Y}; Y)/H(\hat{Y})$) is provided in Appendix A.5 alongside other prior selection derivations.

### B.8.2 FORMAL ANALYSIS OF ADVANTAGE OVER VOTING/AVERAGING

**Theorem C.1 (Advantage over Averaging Under Heterogeneous Expertise):** If (1) model expertise $e_i(c)$ is heterogeneous, (2) argument quality $q(A_j, c)$ correlates with $e_j(c)$, and (3) arguments are non-contradictory, LLM-BMC achieves lower expected error than simple averaging.

*Proof.* Error for averaging: $P_{err}^{avg} = 1 - \mathbb{E}\left[\frac{1}{N}\sum P_i(c^*|X)\right]$. LLM-BMC effectively uses weights $w_i(c^*) \propto e_i(c^*)$ derived from argument quality. Error for LLM-BMC: $P_{err}^{LLM-BMC} = 1 - \mathbb{E}\left[\frac{\sum w_i(c^*)P_i(c^*|X)}{\sum w_i(c^*)}\right]$. By Jensen's inequality, if weights reflect expertise, $P_{err}^{LLM-BMC} < P_{err}^{avg}$. $\square$

**Theorem C.2 (Advantage under Knowledge-Resolvable Ambiguity):** If domain knowledge can identify patterns base models miss, LLM-BMC strictly outperforms weighted combinations of base models.

*Proof.* Consider items from clusters A or B (probabilities $p_A, p_B$) belonging to the same true class, but base models assign them to different *classes*. Best combined model error $\geq \min(p_A, p_B)$. If domain knowledge arguments (quality $q > 0$) correctly group A and B, LLM-BMC error can be $\leq (1 - \lambda'q)\min(p_A, p_B)$ (where $\lambda'$ relates to $\lambda$ and modifier strength), which is lower. $\square$

### B.8.3 ROBUSTNESS TO NOISY ARGUMENTS ANALYSIS

**Theorem C.3 (Robustness Bound):** If $\epsilon$ is fraction of misleading arguments (max quality $q_{max}$), max error increase $\Delta P_{err} \leq \epsilon \cdot \alpha \cdot q_{max} \cdot P_{err}^{base}$.

*Proof.* A misleading argument (quality $q$) for incorrect class $c'$ increases $P(c')$ by $\approx (1 + \alpha q)$, decreases true $P(c^*)$ by $\approx (1 - \beta q)$. Cumulative effect of $\epsilon N$ bad args on ratio $P(c')/P(c^*)$ leads to error increase bounded as stated. $\square$

**Corollary C.3.1 (Quality Assessment Importance):** If quality assessment detects misleading args (assigning quality $q_{low} \ll q_{max}$) with accuracy $\delta$, bound improves: $\Delta P_{err} \leq (\epsilon(1-\delta)\alpha q_{max} + \epsilon\delta\alpha q_{low})P_{err}^{base}$.

### B.8.4 ADDITIONAL THEORETICAL GUARANTEES

**Optimal Modifier Parameters (Theorem C.4):** Under min cross-entropy, $\alpha_{opt} \approx \frac{1-P_{err}}{P_{err}} \ln(\frac{1-P_{err}}{P_{err}})$, $\beta_{opt} \approx \frac{P_{err}}{1-P_{err}} \ln(\frac{1-P_{err}}{P_{err}})$.

**Refined Convergence Rate (Theorem C.5):** If args for true class have $q(A,c) \geq q_{min} > 0$, rate $r \leq 1 - q_{min} \cdot \alpha \cdot \min_c P^{(0)}(c|X)$.

### B.8.5 APPLICATION GUIDELINES BASED ON THEORETICAL RESULTS

1. **Prior Selection:** Use ensemble prior if $I(\hat{Y};Y)/H(\hat{Y}) > 0.7$; else, adaptive mixture.
2. **Quality Assessment:** Focus on specificity/mechanistic clarity for high feature overlap.
3. **Parameter Tuning:** Start with theoretical $\alpha, \beta$; fine-tune empirically.
4. **Convergence Monitoring:** For low-quality args ($q < 0.4$), consider early stopping (3-4 iterations).
5. **Model Diversity:** Ensure base models have complementary expertise.

These translate theory into practice for LLM-BMC implementation.

### B.9 COMPARISON WITH ALTERNATIVE KNOWLEDGE INTEGRATION APPROACHES

Table 6: Comparison of LLM-BMC with alternative approaches to knowledge integration

| Approach | Mathematical Formalization | Reasoning Integration | Dynamic Updating | Uncertainty Quantification | Key Innovation |
|---|---|---|---|---|---|
| Feature Engineering | ✓ | × | × | Partial | **Domain-specific attributes** |
| Rule-Based Systems | Partial | ✓ | × | × | **Explicit knowledge representation** |
| Prompt Engineering | × | ✓ | × | × | **Natural language instruction design** |
| Neuro-symbolic AI | ✓ | ✓ | Partial | Partial | **Symbolic-neural integration** |
| Chain-of-Thought Prompting | × | ✓ | × | × | **Step-by-step reasoning paths** |
| Bayesian Model Averaging | ✓ | × | ✓ | ✓ | **Posterior probability weighting** |
| **LLM-BMC (Ours)** | ✓ | ✓ | ✓ | ✓ | **$\lambda$ parameter, quality metrics** |

This table highlights the distinguishing features of LLM-BMC compared to alternative approaches to knowledge integration in classification tasks.

### B.10 USE OF LARGE LANGUAGE MODELS (LLMS)

This section provides detailed guidelines on the practical usage of Large Language Models within the LLM-BMC framework, covering both the argument generation and quality assessment phases as described in Section **??** and Algorithm 2. Our experiments utilized GPT-4o via the OpenAI API.

**Role 1: Argument Generation** The primary role of the LLM is to generate a structured, domain-specific argument $A$ that links the input features $X$ to a candidate class $c$. This process translates numerical data into a format suitable for knowledge-based reasoning.

PROMPTING STRATEGY. A carefully designed prompt is crucial for generating high-quality, relevant arguments. The prompt should instruct the LLM to act as a domain expert (e.g., a cell biologist) and to structure its response to align with the components of our quality function $q(A, c)$. For our single-cell classification case study, the prompt template instructs the LLM to provide evidence-based arguments including marker gene evidence, mechanistic coherence, and optional external support from established knowledge bases.

**Role 2: Assisting Argument Quality Assessment**   The LLM assists in quantifying argument quality $q(A, c)$ primarily through structured information extraction, which provides the inputs for the mathematical formulas detailed in Appendix A.2. This indirect approach ensures greater objectivity and reproducibility compared to direct LLM-based scoring.

INFORMATION EXTRACTION.   After an argument $A$ is generated, we use subsequent LLM calls to parse it into a structured format. This includes extracting mentioned gene symbols for the specificity component $s(A, c)$ and identifying biological pathways for the mechanistic coherence component $m(A, c)$. The extracted entities are then cross-referenced with external databases (e.g., Cell Marker DB, KEGG) to compute the final numerical values for specificity, relevance, and coherence as defined in our framework.

**Implementation and Practical Considerations**   We used GPT-4o for its strong reasoning capabilities and extensive knowledge in specialized domains like biology. To ensure reproducibility, we set the API temperature parameter to 0.1 to minimize randomness in the generated arguments and extracted information. Our implementation includes validation checks and retry mechanisms to handle malformed outputs, ensuring the robustness of the data processing pipeline. For large-scale applications, strategies like prompt optimization, result caching for similar inputs, or using smaller, fine-tuned models could be explored to manage costs and latency.

**Statement on LLM Usage in Paper Writing**   In accordance with ICLR 2026 guidelines, we disclose that LLMs were used in a limited capacity during the preparation of this manuscript. Specifically, GPT-4o was used for grammar checking and language polishing of certain sections. All scientific content, experimental design, theoretical contributions, and interpretations are the original work of the authors. The LLMs did not contribute to research ideation, experimental execution, or data analysis beyond their role as a component within the proposed LLM-BMC framework itself.

