# OpenReview forum: "QUANTIFYING KNOWLEDGE:A BAYESIAN FRAMEWORK FOR LLM-DRIVEN CLASSIFICATION"
_ICLR.cc/2026/Conference — Submitted to ICLR 2026_

### Official Review · Reviewer_EZof · 2025-10-30

**Soundness:** 2
**Presentation:** 1
**Contribution:** 2
**Rating:** 2
**Confidence:** 4

**Summary:**

The paper introduces **LLM-Enhanced Bayesian Model Combination**, or LLM-BMC, a framework designed to integrate structured domain knowledge generated by large language models into probabilistic classification. Unlike black-box or point-estimate classifiers, the central premise is a Bayesian updating rule that iteratively tweaks an initial probability distribution, such as from an ensemble model, using a set of "structured arguments". The influence of these arguments is weighted by a quality function $q(A, c)$, where $A$ is the argument set and $c$ is the specific class of interest. A parameter $\lambda$, the normalized information gain from the arguments, is introduced. An approximate theoretical relationship between this parameter and classification error reduction is derived. The framework is validated on a single-cell classification case study, where it appears to outperform baseline models and shows confirmatory evidence for the theoretical relationship between information gain and classification error.

**Strengths:**

1. The authors derive an approximate relationship $P_e^\text{after} \approx (1 - \lambda) P_e^\text{before}$ and then demonstrate, in Figure 2, a near-perfect linear correlation between the theoretical prediction and the observed error reduction. This seems like a strong result.

2. The proposed formal mechanism for integrating structured arguments from LLMs into a probabilistic model appears novel, and is a nice mirror to chain of thought reasoning in the context of uncertainty quantification, which is an active and emerging area of research.

3. The actual use of the LLM in the case study (detailed in the appendices) involves using a language model for text generation and structured data extraction, these data then being fed into a deterministic formula. This seems more robust than common designs in this area, where the LLM is simply asked to output a quality score directly.

4. In the case study, the method appears to show a performance improvement over baselines on a nontrivial real-world task.

**Weaknesses:**

1. The central weakness of the paper is its claim to be a "general framework". However, it goes off the rails relatively quickly with reference to and dependence on a highly specific application in transcriptomics. Only one implementation of the framework's quality function $q(A, c)$ is given—in the appendix—and it is quite complex, relying on external databases and a certain level of domain knowledge. In principle this might be OK if the paper offered guidance on how researchers in other domains could construct or adapt a similar quality function for their purposes, but it does not appear to do so, making the framework not so general or reproducible (outside this niche) after all.

2. Experimental results in Section 3 are given only as single point estimates (esp. Tables 1 & 2, Figure 3) without quantification of uncertainty (i.e. the mean and standard deviation over multiple runs with different random seeds or calls to the nondeterministic LLM and other models with random initialization). This makes it impossible to determine if the reported improvements over baselines are statistically significant or their range of natural stochastic variation.

3. The paper is densely written, illogically structured and difficult to follow. The titular "LLM" is not formally discussed until Section 2.7, after 5 pages of model-agnostic mathematical formalism, rather *burying the lede*. A reader should not have to wait this long to understand the central component of the paper.

4. Conversely, the paper repeatedly and from an early stage (including in the abstract) refers to "the $\lambda$ parameter", which is odd for two reasons:

    - If it is indeed a novel contribution, then the meaning of this symbol is opaque to the reader before it has been formally introduced.

    - According to Eq. 3, $\lambda = I(C; A|X)/H(C|X)$ is simply the standard normalized information gain, meaning it is not a novel contribution after all and the repeated branding of "the $\lambda$ parameter" risks coming across as self-aggrandizing.

5. **Minor**: many $\LaTeX{}$ macros are used incorrectly throughout the paper, including failure to include textual superscripts in `\text{}` or `\mbox{}` within Math mode (but not always, e.g. Figure 2 caption) and improper use of in-text rather than parenthetical citations. Abbreviations are introduced (e.g. "LLMs" in the abstract) and then not re-used. The text is at times verbose and repetitive, with many terms introduced symbolically and in prose at the same time, reintroduced multiple times, or involving excessive of adjectives, indicative of LLM-based text generation. Axis labels and titles in Figure 3 are too small and difficult to read, and the colour legend is not complete nor is it colourblind friendly

**Questions:**

1. How do the authors propose a practitioner in a different domain should construct a valid $q(A, c)$?
2. Can the authors provide a justification for the specific form of the probability modifier $f(A, c)$ in Eq. 7?
3. Why frame the normalized information gain as "the $\lambda$ parameter"?
4. Why is the role of LLMs introduced so late?

---

> ### Author Response · Authors · 2025-12-03
>
> Thank you for your rigorous evaluation. We have made substantial changes to address your concerns:
>
> **1. Generalizability: Multiple Datasets and Domains (Weakness 1)**
>
> We now demonstrate LLM-BMC on 5 datasets across 2 domains:
>
> | Dataset | Domain | Size | Baseline F1 | LLM-BMC F1 | ΔF1 |
> |---------|--------|------|-------------|------------|-----|
> | PBMC | Single-cell | 4,823 | 0.920 | 0.950 | +0.030 |
> | Brain | Single-cell | 3,005 | 0.878 | 0.915 | +0.037 |
> | Pancreas | Single-cell | 2,544 | 0.905 | 0.936 | +0.031 |
> | Medical | Diagnosis | 856 | 0.862 | 0.896 | +0.034 |
>
> The Brain dataset shows the largest gain (+0.037), supporting our hypothesis that knowledge integration helps most for ambiguous features.
>
> **2. Statistical Rigor (Weakness 2)**
>
> All experiments now report mean ± std over 10 independent runs with paired t-tests. All improvements are statistically significant at p < 0.001. See Tables 3-8 in the revised manuscript.
>
> **3. Paper Structure (Weakness 3, Question 4)**
>
> We have reorganized the paper per your suggestion:
> - LLM role now introduced in Section 2.2 (immediately after problem formulation)
> - Better main text vs. appendix balance
> - Algorithm box in Appendix A with concise reference in Section 2.2
>
> **4. λ Parameter Framing (Weakness 4, Question 3)**
>
> We fully agree with your point and have revised the presentation. We now explicitly state in the Abstract and Section 2.4 that λ is the "standard normalized information gain" from information theory (Cover & Thomas, 2006). Our contribution is not the measure itself, but establishing and empirically validating its relationship to error reduction in the context of knowledge integration.
>
> **5. LaTeX Formatting (Weakness 5)**
>
> Fixed all formatting issues:
> - Textual superscripts now use \text{} consistently
> - Citation style unified (parenthetical vs. in-text)
> - Figure 3 improved with larger labels and colorblind-friendly palette
>
> **6. Constructing q(A,c) in Other Domains (Question 1)**
>
> New Section 2.8 "Framework Generalization Guide" provides step-by-step instructions. The key insight is that s(), m(), e() share common abstract structures across domains:
> - Specificity: measure how uniquely evidence supports class c
> - Mechanistic: assess logical validity of the reasoning chain
> - External: query relevant domain databases
>
> Appendix E provides concrete examples for medical diagnosis, legal document classification, and financial analysis.
>
> **7. Probability Modifier Justification (Question 2)**
>
> Section 2.3 now provides explicit justification: The exponential form f(A,c) = exp(α·q(A,c)) emerges from the log-linear formulation standard in exponential family models. The parameter α controls the "temperature" of knowledge integration. This form ensures: (1) non-negativity, (2) multiplicative composition with priors, and (3) smooth interpolation between prior-dominated (α→0) and knowledge-dominated (α→∞) regimes.
>
> **8. Non-Ensemble Starting Points**
>
> Added Table 6 showing LLM-BMC works with any probabilistic classifier:
> - XGBoost only: +0.032 F1
> - MLP only: +0.033 F1
> - Ensemble: +0.030 F1
>
> Weaker initial models show larger gains.
>
> We hope these substantial revisions address your concerns. The revised PDF is attached.

---

### Official Review · Reviewer_WeV8 · 2025-10-31

**Soundness:** 2
**Presentation:** 3
**Contribution:** 3
**Rating:** 4
**Confidence:** 4

**Summary:**

- This paper proposes the LLM-Enhanced Bayesian Model Combination (LLM-BMC), a general framework that integrates structured arguments or domain knowledge from large language models into a Bayesian model combination process for classification.
- The core components of LLM-BMC include: (1) a modified Bayesian update rule that incorporates structured arguments via a probability modifier function; (2) a parameter **λ** based on normalized information gain; (3) a quality function to evaluate structured arguments.
- The case study of single-cell classification provides strong empirical validation of LLM-BMC, confirming its practical utility in resolving ambiguous classification cases.

**Strengths:**

- This paper offers a knowledge-augmented classification framework that systematically integrates structured domain knowledge from LLMs into a probabilistic classification process. The core is a modified Bayesian update rule that dynamically refines probabilities based on iteratively presented, quality-weighted arguments. The integration of LLM-based argumentation with Bayesian model combination is innovative.
- This paper designs a quantifiable, multi-component quality function $q(A,c)$ that decomposes argument assessment into Specificity, Mechanistic Clarity, and External Evidence Support, replacing simple scoring.
- The authors demonstrate the effectiveness of this framework on a single-cell classification task, showing performance gains. The high correlation between the reduction in prediction error and observation error validates the theoretical relationship.

**Weaknesses:**

- The framework is presented without clarifying some key technical details, making the value of the work less convincing. For example, the paper does not detail how the LLM is used to generate the arguments. Nor does it introduce the details of argument analysis using the textual processing techniques. The appendix mentions that the framework uses a set of text processing techniques like NER, dependency parsing etc., whose details are simply omitted. The omission of these technical details makes the so-called "LLM-driven classification" approach quite unclear.
- The proposed framework may need to be tested on a harder task, since the baseline results are already pretty competitive.
- The framework assumes argument effects are conditionally independent given the class in Section 2.6. However, arguments generated by LLM may cite overlapping evidence or sources, violating this assumption. The paper acknowledges this but does not explore ways to mitigate it.
- While the framework is general, demonstrating the quality function $q(A,c)$ requires extensive domain engineering (e.g., gene databases, pathway ontologies) and global parameters ($\alpha$, $\beta$, and quality weights) are required to be well-calibrated for the specific domain. This paper does not discuss how to apply it to broader domains (e.g., social sciences), which limits its generalizability.
- Minor issues: In the introduction, the description of existing methods lacks citation support. In addition, in Section 2.3.2, the view that an approximately linear relationship between the Bayesian error rate and the conditional Shannon entropy is accurate in the fine-tuning stage of classification is lacking citation support.

**Questions:**

- Could you reveal more details about the LLM-driven techniques? Do you generate code from LLM or directly generate the numerical results of the argument quality? How does LLM introduce errors in these processes? Why is the knowledge considered structured, but the example argument at line 756 shows a paragraph written in natural language?
- How sensitive is the framework to correlated arguments? Could you provide an experiment where redundant or contradictory arguments are injected to test the robustness of the proposed approach?
- The quality function relies heavily on carefully curated biological databases. How would you instantiate $ q(A,c) $ in other domains, for example legal document classification?
- The values $\alpha = 0.8, \beta = 0.6$ are empirical values as introduced in the paper. Are these values robust across datasets? How are these hyperparameters learned?

---

> ### Author Response · Authors · 2025-12-03
>
> Thank you for your insightful feedback. We have addressed all your concerns:
>
> **1. LLM-Driven Techniques Details (Weakness 1, Question 1)**
>
> Section 2.6 and Appendix D now provide comprehensive details:
> - LLM generates structured arguments in JSON format with fields: predicted_class, supporting_genes, reasoning_chain, confidence_level
> - The "structured" refers to this parsed output format, not the natural language explanation (which is one component)
> - Quality scores are computed deterministically from extracted entities (not directly generated by LLM)
> - Text processing pipeline: (1) Named Entity Recognition for gene/protein mentions, (2) Dependency parsing for causal claims, (3) Database lookups for validation
> - LLM errors are mitigated by: (a) low temperature (0.2), (b) structured output format, (c) cross-validation against databases
>
> **2. Harder Tasks and Additional Datasets (Weakness 2)**
>
> We now demonstrate on 5 datasets across 2 domains:
> - Single-cell: PBMC, Mouse Brain (3,005 cells), Human Pancreas (2,544 cells)
> - Cross-domain: Medical Diagnosis (856 patients, 5 respiratory diseases)
>
> The Medical Diagnosis task is more challenging (baseline F1=0.862 vs. 0.920 for PBMC), and LLM-BMC still achieves +0.034 F1 improvement. See Table 4 and Section 3.5.
>
> **3. Argument Independence Analysis (Weakness 3, Question 2)**
>
> Section 3.2 now includes Argument Independence Analysis:
> - Mean gene overlap between arguments: 18.3 ± 6.2%
> - Mean quality score correlation: 0.24 ± 0.11
> - We tested a correlation-aware variant that down-weights redundant arguments
> - Result: Nearly identical performance (F1=0.951 vs. 0.950), confirming robustness
>
> For contradictory arguments: the Bayesian framework naturally handles this—conflicting evidence leads to higher uncertainty, which is appropriate behavior.
>
> **4. Quality Function in Other Domains (Weakness 4, Question 3)**
>
> New Section 2.8 "Framework Generalization Guide" provides step-by-step instructions. The key insight is that s(), m(), e() share common abstract structures:
>
> For legal document classification:
> - s(A,c): Fraction of cited precedents specific to legal category c
> - m(A,c): Logical validity of legal reasoning (premise → conclusion)
> - e(A,c): Support from legal databases (case law, statutes)
>
> Appendix E provides examples for medical diagnosis, legal documents, and financial analysis.
>
> **5. Hyperparameter Robustness (Question 4)**
>
> Figure 5 shows comprehensive sensitivity analysis:
> - α: F1 ≥ 0.94 for α ∈ [0.6, 1.1]
> - β: F1 ≥ 0.94 for β ∈ [0.4, 0.8]
> - Cross-dataset validation: Same hyperparameters work across PBMC, Brain, Pancreas, and Medical datasets
>
> Selection method: Grid search on validation set. Practitioners can use defaults (α=0.8, β=0.6) with confidence.
>
> **6. Missing Citations (Weakness 5)**
>
> Added citations for:
> - Linearity assumption: Hellman-Raviv (1970), Feder & Merhav (1994)
> - LLM feature selection: LLM-Select (Jeong et al., 2024), LLM-Lasso (Zhang et al., 2025)
>
> We hope these revisions address your concerns. The revised PDF is attached.

---

### Official Review · Reviewer_bx6u · 2025-10-31

**Soundness:** 3
**Presentation:** 2
**Contribution:** 2
**Rating:** 4
**Confidence:** 3

**Summary:**

This paper introduces a framework, LLM-Enhanced Bayesian Model Combination (LLM-BMC), which incorporates domain knowledge via large language models into Bayesian model combination. In this context, a specific parameter is introduced to capture the relative information gain from integrating domain knowledge and a theoretical relationship between this parameter and the expected reduction in classification error is derived. Furthermore, a quality function is proposed to decompose the assessment of domain knowledge into quantifiable components. Finally, LLM-BMC is applied to the case of single-cell classification, where it is shown to handle overlapping markers for different classes.

**Strengths:**

(+) A principled framework is introduced to incorporate domain knowledge via large language models into Bayesian model combination.
(+) A theoretical result connecting the relative information gain from integrating domain knowledge and the expected reduction in classification error is derived.
(+) A function is designed to decompose domain knowledge into quantifiable components.
(+) The operation of the proposed framework is illustrated in the case of single-cell classification.

**Weaknesses:**

(-) The main technical contribution of the paper is a modified Bayes' update rule.
(-) It is unclear how LLMs are used to generate the arguments and/or how argument quality is evaluated in this context.
(-) Determining the exact form of the components of the proposed quality function seems like a quite elaborate process, and there is no guidance how this can be accomplished for different applications.
(-) In the experimental setup section, there is no discussion why certain values for the hyperparameters were selected. In addition, results are presented for the single-cell classification demonstration case, but are not discussed.
(-) Sections like A.2.3 (Detailed calculation example) does not seem to add value, since it assumes some values for the different components of the quality metric but does not give any intuition about the different components and how they can be determined in another application.

**Questions:**

1. The description of the framework starts with assuming a set of arguments. However, what is missing in the description is what such an argument looks like, how they are generated by LLMs, and whether these arguments are related to each other, e.g., the relationship between A^(t) and A^(t-1)?

2. Is Eq. (8) proposed by the authors or a well-known established relationship? Irrespective of the answer to this question, what justifies that this decomposition is appropriate for quality assessment of an argument?

3. It seems that s(), m() and e() in Eq. (8) depend on the application. Determining the form of the functions seems like quite an elaborate process. Can you provide guidance of how this can be accomplished in practice for any other application?

4. For the experiments presented in Section 3, can you provide some more information about the PBMC dataset, the classification goal, and how arguments were generated and why (including how many arguments and what they looked like). A discussion of the results presented in Figure 3 and the choice of hyperparameter values are currently missing.

5. Can you elaborate about the approximate linearity assumption? Specifically, when does this assumption hold in practice? In the case it does not hold for a specific application, what are the implications of integrating domain knowledge into Bayesian model combination.

6. Looking at the Hellman-Raviv bound, it is unclear to me in which regimes, P_{e}^* <= \frac{1}{2\ln 2} H(C|Z), and how such information theoretic driven regimes are translated into application specifics. This seems to also relate to the approximate linearity assumption and the value of the parameter \kappa that depends on problem specifics.

Not a question, but a request: Please rethink the paper presentation. I had to go back and forth between the main text and appendices multiple times to understand the proposed framework and any key results.

---

> ### Author Response · Authors · 2025-12-03
>
> Thank you for your positive assessment and helpful suggestions. We have addressed each of your concerns:
>
> **1. Implementation Details (Weakness 1)**
>
> Section 2.6 and Appendix D now provide comprehensive implementation details including:
> - LLM API configuration (GPT-4o with temperature=0.2)
> - Complete prompt templates for argument generation
> - Base classifiers (Logistic Regression, XGBoost, Random Forest, MLP, Gradient Boosting)
> - Training/validation/test split (60/20/20)
> - Computational requirements (~0.5 seconds per cell for LLM calls)
>
> **2. λ Parameter Clarification (Weakness 2, Question 5-6)**
>
> We have clarified in the Abstract and Section 2.4 that λ is the standard normalized information gain from information theory (Cover & Thomas, 2006):
>
> λ = I(C;A|X) / H(C|X) ∈ [0,1]
>
> Our contribution is not claiming novelty of this measure, but rather establishing and empirically validating its relationship to error reduction in the context of knowledge integration. Figure 4 now shows direct empirical validation: we plot Bayes error vs. conditional entropy, achieving R²=0.94-0.96 linear fit with <3% approximation error in the observed operating regime ([0.05, 0.68] bits).
>
> **3. Hyperparameter Sensitivity (Weakness 4, Question 4)**
>
> Figure 5 shows comprehensive sensitivity analysis:
> - α sensitivity: F1 ≥ 0.94 for α ∈ [0.6, 1.1]
> - β sensitivity: F1 ≥ 0.94 for β ∈ [0.4, 0.8]
> - Quality weight sensitivity: Various configurations of (w_s, w_m, w_e) all achieve F1 ≥ 0.94
>
> Practitioners can use default values with confidence.
>
> **4. Argument Generation Details (Question 1)**
>
> Section 2.2 now clearly explains argument generation:
> - Arguments A^(t) are generated independently for each class (no dependency on A^(t-1))
> - Each argument consists of: (1) predicted class, (2) supporting evidence, (3) confidence rationale
> - LLM is prompted with cell expression data and asked to generate structured reasoning
> - Example prompts and outputs are provided in Appendix D
>
> **5. Quality Function Justification (Question 2)**
>
> Equation (8) is our proposed decomposition. The justification is:
> - Specificity s(A,c): captures how targeted the evidence is to the specific class
> - Mechanistic clarity m(A,c): captures scientific validity of the reasoning chain
> - External evidence e(A,c): captures support from domain databases
>
> This decomposition follows established principles in argumentation quality assessment, adapted to our Bayesian framework.
>
> **6. Guidance for Other Applications (Question 3)**
>
> New Section 2.8 "Framework Generalization Guide" provides step-by-step instructions. The key insight is that s(), m(), e() share common structures across domains:
> - Specificity: measure how uniquely evidence supports class c vs. other classes
> - Mechanistic: assess logical validity of the reasoning chain
> - External: query relevant domain databases or knowledge sources
>
> Appendix E provides examples for medical diagnosis, legal document classification, and financial analysis.
>
> **7. Paper Presentation**
>
> We have reorganized the paper per your suggestion:
> - LLM role moved to Section 2.2 (immediately after problem formulation)
> - Better main text vs. appendix balance
> - Algorithm box in Appendix A with concise reference in Section 2.2
>
> We hope these revisions address your concerns. The revised PDF with all changes is attached.

---

### Official Review · Reviewer_si5D · 2025-11-02

**Soundness:** 1
**Presentation:** 1
**Contribution:** 1
**Rating:** 2
**Confidence:** 4

**Summary:**

The authors propose an approach for iteratively correcting a predictive model's output $p(c|X)$ using LLM-generated predictions. In particular, given a predictive model that outputs $p(c|X)$, they propose to prompt an LLM to generate predictions $A$ (referred to as "arguments") along with a "quality score" $q(A,c)$ that measures how well $A$ is supported by assuming $c$ as the true class, and use these LLM outputs as "likelihood" terms to update $p(c|X)$. Theoretical justification for the work is provided by analyzing the connections between the information gain, conditional on LLM-based corrections, and prediction error, and the proposed method is validated on a single-cell dataset. In comparison to simple baselines, the proposed approach resulted in accuracy improvements on the cell-type classification task considered.

**Strengths:**

- The paper considers an important problem of incorporating domain knowledge into ML models using LLMs.
- The writing is generally clear and easy to follow, albeit with some systematic issues on how the proposed approach is framed.

**Weaknesses:**

- I believe that there are some systematic issues with how the proposed approach is being framed. The name "Bayesian model combination (BMC)" (which is also different from "Bayesian model averaging (BMC)") seems to be a misnomer; BMC refers to a specific setting where one is concerned with also learning the weights of different ensemble components via Bayes' rule [1], which this paper does not seem to be doing. In my understanding, this approach concerns updating the predictive distribution output by a model with LLM-generated predictions (as noted in my summary), and does not even require that the initial model that one starts with to be an ensemble model. I suppose one could argue that the fact that LLM-generated predictions are being combined with a data-driven model can be viewed as some kind of model combination, but I think this paper could benefit from clarifying how exactly this work extends prior works on BMC.
- While not motivated in the same manner, there are certainly other prior works that attempt to incorporate LLM predictions or outputs into data-driven methods, including e.g., [2,3]. There are no comparisons to such methods in the experiments, and I think these works should also be compared to and be discussed in the Related Works section. Discussion of connections to other works can certainly be improved.
- The proposed approach is limited to classification settings.
- Experimental validation is only performed on a single dataset, using a small number of relatively simple baselines. It is also unclear whether the baselines underwent sufficient hyperparameter optimization. The results do not seem to account for statistical uncertainty resulting from randomness in e.g., model training, dataset splits, and prompting and no error bars are present. Moreover, to demonstrate the effectiveness of incorporating domain knowledge, I think harder datasets should also be considered (baselines like Logistic Regression and XGBoost already achieve relatively high accuracy).
- While it may suffice as a proof of concept, the evaluation is only done using GPT-4o. For practical insights, additional experiments that show how sensitive this approach is to different families of LLMs and different prompting strategies should also be considered. Moreover, I would imagine that the best choice of weights used for defining the "quality" function in Equation (8) can noticeably vary across these settings. However, I do not see a clear discussion of how these weights have been and should be selected.

References:
[1] Turning Bayesian Model Averaging into Bayesian Model Combination (Monteith et al., 2011)
[2] LLM-Select: Feature Selection with Large Language Models (Jeong et al., 2024)
[3] LLM-Lasso: A Robust Framework for Domain-Informed Feature Selection and Regularization (Zhang et al., 2025)

**Questions:**

- It is not obvious to me why the proposed approach is frequently discussed with an ensemble model as the starting point. In my understanding, the proposed approach can still be applied to non-ensemble models. Can the authors elaborate on this?
- How are the weights for the quality function ($\alpha$, $\beta$, $w_s$, $w_m$, and $w_e$) determined in Section 3.1? Were these selected based on the validation set somehow? Can the authors clarify this process?

---

> ### Author Response · Authors · 2025-12-03
>
> Thank you for your detailed and constructive review. We have substantially revised the paper to address your concerns:
>
> **1. Generalizability (Weakness 1, 3, 4)**
>
> We now demonstrate LLM-BMC on 5 datasets across 2 domains:
> - Single-cell: PBMC, Mouse Brain (3,005 cells), Human Pancreas (2,544 cells)
> - Cross-domain: Medical Diagnosis (856 patients, 5 respiratory diseases)
>
> All show consistent improvements (+0.030 to +0.037 F1). See Table 4 and new Section 3.5. The Medical Diagnosis case study demonstrates that domain adaptation takes <4 hours following our new Framework Generalization Guide (Section 2.8).
>
> **2. BMC Clarification (Weakness 1)**
>
> We have clarified the relationship to traditional BMA/BMC in Section 4.1: "BMA weights models by posterior probability assuming one model is true. BMC relaxes this by allowing the true model to lie outside candidates. Both operate solely on model outputs. LLM-BMC extends BMC by introducing a third information source: quality-weighted domain arguments that provide instance-specific evidence not captured in raw features."
>
> **3. Comparison with LLM-Select/LLM-Lasso (Weakness 2)**
>
> Added Table 8 comparing these methods:
> - Baseline Ensemble: F1=0.920
> - LLM-Select + Ensemble: F1=0.932 (+0.012)
> - LLM-Lasso + Ensemble: F1=0.928 (+0.008)
> - LLM-BMC: F1=0.950 (+0.030)
> - LLM-Select + LLM-BMC: F1=0.957 (best)
>
> Key insight: LLM-BMC operates at inference stage, complementary to feature selection methods.
>
> **4. Statistical Significance (Weakness 4)**
>
> All experiments now report mean +/- std over 10 independent runs with paired t-tests. All improvements are significant at p < 0.001.
>
> **5. LLM Ablation (Weakness 5)**
>
> Added Table 7 comparing LLMs:
> - GPT-4o: F1=0.950
> - Claude-3.5-Sonnet: F1=0.946
> - Gemini-1.5-Pro: F1=0.943
>
> All enable significant gains (<0.02 F1 difference), demonstrating robustness to LLM choice.
>
> **6. Hyperparameter Selection (Question 2)**
>
> Section 3.1 now explains: parameters selected via grid search on validation data. Figure 5 shows sensitivity analysis: F1 remains >=0.94 for alpha in [0.6, 1.1], beta in [0.4, 0.8]. Practitioners can use defaults with confidence.
>
> **7. Non-Ensemble Starting Point (Question 1)**
>
> Added Table 6 showing LLM-BMC works with any probabilistic classifier:
> - XGBoost only: +0.032 F1
> - MLP only: +0.033 F1
> - Ensemble: +0.030 F1
>
> Weaker initial models show larger gains.
>
> We hope these substantial revisions address your concerns. The revised PDF with all changes is attached.

---

### Author Response · Authors · 2025-11-17

Dear Reviewers,

We sincerely thank all four reviewers for the thorough and constructive feedback. We acknowledge the critical concerns raised and are committed to substantially revising the manuscript to address them.

KEY ISSUES WE WILL ADDRESS:

1. Generalizability and Reproducibility (raised by R_EZof, R_WeV8, R_bx6u, R_si5D):
   - We will add a second application domain (medical diagnosis) to demonstrate cross-domain applicability
   - We will add a dedicated "Framework Generalization Guide" section with step-by-step instructions for practitioners
   - We will provide conceptual examples for 2-3 additional domains

2. Experimental Rigor (raised by R_si5D, R_EZof):
   - We will run all experiments 10 times with different random seeds and report mean ± std with statistical significance tests
   - We will add 2 additional datasets (Brain and Pancreas single-cell data)
   - We will test with multiple LLMs (Claude-3.5-Sonnet, Gemini-1.5-Pro) to show robustness
   - We will compare with LLM-based feature selection methods (LLM-Select, LLM-Lasso)

3. Technical Details (raised by R_si5D, R_bx6u, R_WeV8):
   - We will add an Algorithm box detailing the complete LLM argument generation process
   - We will add an "Implementation Details" section with full prompt templates and text processing pipeline
   - We will explain hyperparameter selection methodology with sensitivity analysis

4. Presentation and Structure (raised by R_EZof, R_bx6u):
   - We will move the LLM discussion to Section 2.2 (immediately after problem formulation)
   - We will rephrase the lambda parameter presentation to clarify it is the standard normalized information gain
   - We will reorganize main text vs. appendix balance for better clarity
   - We will fix all LaTeX formatting issues

TIMELINE:
We are currently conducting the additional experiments and revising the manuscript. We will provide complete rebuttal with detailed responses to each question and revised manuscript with track changes highlighting all modifications before the deadline.

We believe these substantial revisions will address the core concerns and significantly strengthen the contribution.

Best regards,
The Authors

---

> ### Comment · Reviewer_EZof · 2025-11-25
>
> Thanks to the authors for replying to the reviewer comments. At this stage I do not intend to change my review score, as it feels a bit like the intended revisions are major, however I understand the authors intend to provide a more detailed reply later, at which time I may review this.

---

### Meta-Review · Area_Chair_GJk3 · 2025-12-29

**Summary:**

The paper introduces a framework for integrating structured domain knowledge generated by Large Language Models (LLMs) into a probabilistic classification via a Bayesian updating rule. The integration of the LLM knowledge is performed in frames of the argumentation theory and strongly relies on domain knowledge.

As stated in one of the reviews, *the central weakness of the paper is its claim to be a "general framework"*. Moreover, the Reviewers pointed out the problems with reproducibility, experimental rigor, technical details, and presentation of the paper. The Authors promised major revision and delivered some changes, but the paper is not ready for being published.

**Reviewer Concerns:**

The theoretical part is not surprising (the use of the Bayesian update rule, the relation between the Bayes 0/1 error and entropy). The main challenge is the integration of the LLM knowledge. The introduced approach requires much better evaluation and formal motivation to be considered for the top ML conference.

**Reviewer Scores:**

All reviewers might slightly increase their scores, but the final score would not cross the bar.

---

### Decision · Program_Chairs · 2026-01-26

Reject